# Extracellular matrix-driven metabolic control of pancreatic endocrine lineage allocation

Christine Ebeid[1,4,7], Adam Rump [ID] [1,2,5,7], Chenglei Tian [ID] [1,2], Anant Mamidi [ID] [1,6], Adèle De Arcangelis [ID] [3], Gérard Gradwohl [ID] [3] & Henrik Semb [ID] [1,2]✉

## Abstract

The mechanical and metabolic states of progenitor and stem cells are emerging as key regulators of cell fate decisions. Lineage specification of pancreatic endocrine cells is promoted by reduced mechanical tension in vitro, but the underlying mechanism is poorly understood. Here, we show that heterogeneously deposited low-adhesion extracellular matrix (ECM) components, such as the laminin isoform LN411, trigger a local "soft" environment by broadly reducing the expression of integrins. Mimicking this low-tension state by in vitro knockdown and in vivo gene targeting of the LN-binding integrins *Itga3* and *Itga6* reveal their importance in inducing endocrinogenesis. Unexpectedly, the cell responds to this change in tensile forces by engaging a major metabolic enzyme, PDK4, to execute the resulting cell fate decision. PDK4 achieves this through two distinct mechanisms: a non-canonical action controlling YAP activity and a canonical metabolic function maintaining PDX1 expression. In sum, we believe our findings have broad relevance for how local changes in mechanical tension governs cell behaviour in many developmental and disease contexts.

**Keywords** Pancreatic Endocrinogenesis; PDK4; Laminin-411; Mechanotransduction; YAP Signalling
**Subject Categories** Cell Adhesion, Polarity & Cytoskeleton; Development; Metabolism

## Introduction

Recent advances in single-cell RNA sequencing have significantly enhanced our understanding of the gene regulatory networks governing cell fates (Asp et al, 2019; La Manno et al, 2021; Olaniru et al, 2023). However, much less is known about the identity and mechanisms of the environmental cues that control the fate-inducing transcriptional machinery. Here, we unravel the identity and mode of action of important niche cues that control the gene

regulatory apparatus involved in the differentiation of the endocrine cell lineages during pancreas development.

During pancreas development, the epithelium is segregated into a pro-acinar "tip" and a bipotent ductal/endocrine compartment. Endocrine progenitor cells are specified in a salt and pepper pattern within the bipotent epithelium by induction of the transcription factor *Neurog3* (Apelqvist et al, 1999; Gu et al, 2002; Gradwohl et al, 2000). However, our understanding of the identity and functional mechanisms of the extracellular cues that induce differentiation of endocrine progenitors remains incomplete. A lateral inhibition mechanism has been proposed to control *Neurog3* induction, wherein Neurog3 upregulates the Notch ligand Dll1 to repress *Neurog3* expression in neighboring cells (Qu et al, 2013). However, the expression patterns of both Notch ligands and the key Notch target gene *Hes1* are inconsistent with a classical lateral inhibition model (Ahnfelt-Rønne et al, 2012). An alternative model is thus needed to explain how a small, scattered subset of bipotent pancreatic progenitor cells is specified toward the endocrine lineage.

Recently, we and others identified YAP-mediated mechano-signalling as an upstream regulator of Notch signaling during development of the pancreas and a variety of other tissues (Mamidi et al, 2018; Engel-Pizcueta and Pujades, 2021; Cebola et al, 2015). While YAP activity is classically regulated by the Hippo pathway, the mild phenotype of pancreas-specific Hippo component knockout suggests Hippo-independent regulation during pancreas development (Mamidi et al, 2018; Gao et al, 2013). Recent work revealed extensive Hippo-independent regulation of YAP activity by the extracellular matrix (ECM), actin dynamics, and cellular metabolism (Panciera et al, 2017; Chi et al, 2020; Ibar and Irvine, 2020). Consistently, we demonstrated that confinement of pancreatic bipotent progenitors in vitro, for example via certain ECM molecules or actin depolymerization, promotes endocrine specification through reduced YAP activity (Mamidi et al, 2018). However, what remains unclear is whether this machinery is operational in vivo, and by which environmental signal it is controlled.

Recent studies have demonstrated that metabolic pathways play key roles in the regulation of cell fates. For instance, glucose metabolism controls the first lineage segregation during

[1]Novo Nordisk Foundation Center for Stem Cell Biology (DanStem), University of Copenhagen, Copenhagen 2200, Denmark. [2]Institute of Translational Stem Cell Research, Helmholtz Diabetes Center, Helmholtz Zentrum München, Neuherberg 85764, Germany. [3]Institut de Génétique et de Biologie Moléculaire et Cellulaire, CNRS UMR 7104/INSERM U1258/Université de Strasbourg, Illkirch 67404, France. [4]Present address: Cell Therapy R&D, Novo Nordisk A/S, Måløv 2760, Denmark. [5]Present address: Chemometec A/S, Allerød 3450, Denmark. [6]Present address: FUJIFILM Diosynth Biotechnologies, Hillerød 3400, Denmark. [7]These authors contributed equally: Christine Ebeid, Adam Rump. ✉E-mail: henrik.semb@helmholtz-munich.de

mammalian development by regulating YAP nuclear localization through the hexosamine biosynthetic pathway, thereby specifying trophectodermal fate (Chi et al, 2020). This metabolic control extends to major developmental transitions, as demonstrated by mannose metabolism directing mesoderm specification during gastrulation (Cao et al, 2024). However, while these studies establish metabolism as a key regulator of development, both the roles of specific metabolic regulators in organ-specific cell fate decisions and the way in which these regulators are influenced by environmental cues remain largely unexplored.

In this study, the low adhesion LN411 is identified as the key extrinsic regulator of endocrine fate choice. LN411 adhesion leads to reduced expression of integrins, including the laminin-binding integrins α3 and α6, in bipotent progenitor cells. We show that in vitro knockdown and in vivo knockout of these integrins increase endocrine induction. Hence, heterogeneous deposition of LN411 within the trunk epithelium may explain why a subset of bipotent pancreatic progenitor cells adopt an endocrine fate.

Surprisingly, we identify the metabolic regulatory kinase PDK4, which regulates conversion of pyruvate to acetyl-CoA (Holness and Sugden, 2003), as a key mediator of LN411-induced endocrine specification. Specifically, exposure to LN411 induces upregulation of *PDK4*, which induces endocrinogenesis through dual functions. While YAP is inhibited by a non-canonical mechanism, *PDX1* expression is maintained via reduced production of acetyl-CoA. Altogether, our findings provide a new paradigm for how cells integrate mechanosignalling and metabolism to control cell fate decisions.

# Results

## LN411 induces endocrine specification of pancreatic progenitors

We previously showed that plating pancreatic progenitors on human placenta laminin induces their endocrinogenesis (Mamidi et al, 2018). To pinpoint which laminin isoform is most potent in inducing endocrinogenesis in vitro, we reseeded human embryonic stem cell (hESC)-derived pancreatic progenitors from the HUES4 cell line on nine different laminin isoforms. The extent of endocrine induction was notably contingent on the specific laminin isoform, with LN211 and LN411 being the most effective inducers of NEUROG3, insulin, and glucagon expression (Fig. 1A–D). Importantly, while both LN211 and LN411 showed strong endocrinogenic effects in the HUES4 cell line, only the effect of LN411 was conserved in the independent SA121-derived NEU-ROG3 reporter (NGN3-GFP) hESC cell line (Fig. EV1A–C). This cell line-independent effect led us to focus our subsequent mechanistic studies on LN411. Interestingly, the endocrinogenic effect by LN211 and LN411 was associated with cell clustering, whereas cells reseeded on fibronectin and the other laminin isoforms spread to form a monolayer (Figs. 1A and EV1A and EV2A,B). As anticipated from our previous model of laminin-induced endocrine specification, LN411 induced decreased levels of actin polymerization, nuclear YAP, and expression of YAP target genes (Fig. EV2C,D). Collectively, these findings underscore a distinct and specific pro-endocrinologic effect of LN411 in vitro across multiple cell lines.

## LN411 induces endocrinogenesis via reduced cell-ECM adhesion

LN411 has previously been shown to have lower adhesion properties compared to other laminin isoforms (Ishikawa et al, 2014). This decreased adhesion has been shown to impact the surface expression of integrins, as weak binding of integrins to the ECM disrupts focal adhesion and stress fiber formation, leading to integrin internalization (Du et al, 2011). Consistently, we found a coordinated reduction in the expression of both fibronectin-binding integrins α5 and αV and laminin-binding integrins α3 and α6 at both the RNA and protein levels upon exposure to LN411 (Figs. 2A and EV3A). To assess whether the decreased expression of integrins is directly associated with endocrinogenesis, we examined the expression of each integrin in NGN3-GFP⁺ hESCs upon latrunculin B treatment (Mamidi et al, 2018). Indeed, the expression of each of the four integrins was reduced in NGN3-GFP⁺ endocrine progenitors compared to their NGN3-GFP⁻ counterparts (Fig. EV3B,C). These findings suggest that LN411 induces endocrinogenesis via a reduction in cell-ECM adhesion.

To ascertain whether the reduced expression of integrins by LN411 is secondary to increased endocrine induction, we measured the expression of each integrin 3, 6, or 24 h after reseeding. Remarkably, the expression of each integrin was reduced at every time point. At 6 h after reseeding, the expression of *ITGA6*, *ITGA5* and *ITGAV* was significantly reduced, while there was no difference in the expression of any endocrine marker gene at this time point (Fig. 2B). This indicates that reduced expression of integrins precedes endocrine induction upon reseeding on LN411.

To genetically validate the direct role of integrin α3 and α6 in endocrinogenesis, siRNA-based knockdown experiments were performed. Knockdown of either integrin α3 or α6 was sufficient to enhance endocrine specification (Fig. 2C,D). Importantly, this enhancement occurs when cells are cultured on LN521, precluding a specific effect of LN411-integrin interactions. These findings collectively demonstrate that reduced expression of laminin-binding integrins α3 and α6 actively promotes endocrine specification, thus providing compelling evidence for a mechanistic link between LN411-induced reduction in cell-ECM adhesion and the induction of endocrine fate in pancreatic progenitors.

## Reduced expression of laminin-binding integrins promotes endocrinogenesis in vivo

To validate our in vitro findings in vivo, we analyzed the consequences of knocking out integrins α3 and α6 in the developing mouse pancreas. First, we examined the distribution of laminin isoforms in the pancreatic epithelium at E15.5, which revealed a heterogenous distribution of laminin α4 (Fig. EV4). Due to the previously reported functional redundancy between *Itga3* and *Itga6*, we generated single and double knockouts by intercrossing *Itga3*⁺/⁻ and *Itga6*⁺/⁻ mice (De Arcangelis et al, 1999). Given the embryonic lethality of the double knockouts, we conducted the analysis using an ex vivo explant system (Fig. 3A).

Knockout of either *Itga3* or *Itga6* alone had no discernible effects on pancreas size, morphology, or proportion of endocrine cells (Fig. 3D). However, the pancreas of the *Itga3/Itga6* double knockouts exhibited an expansion of the endocrine compartment, coupled with hypoplasia and reduced branching (Fig. 3B–D). A

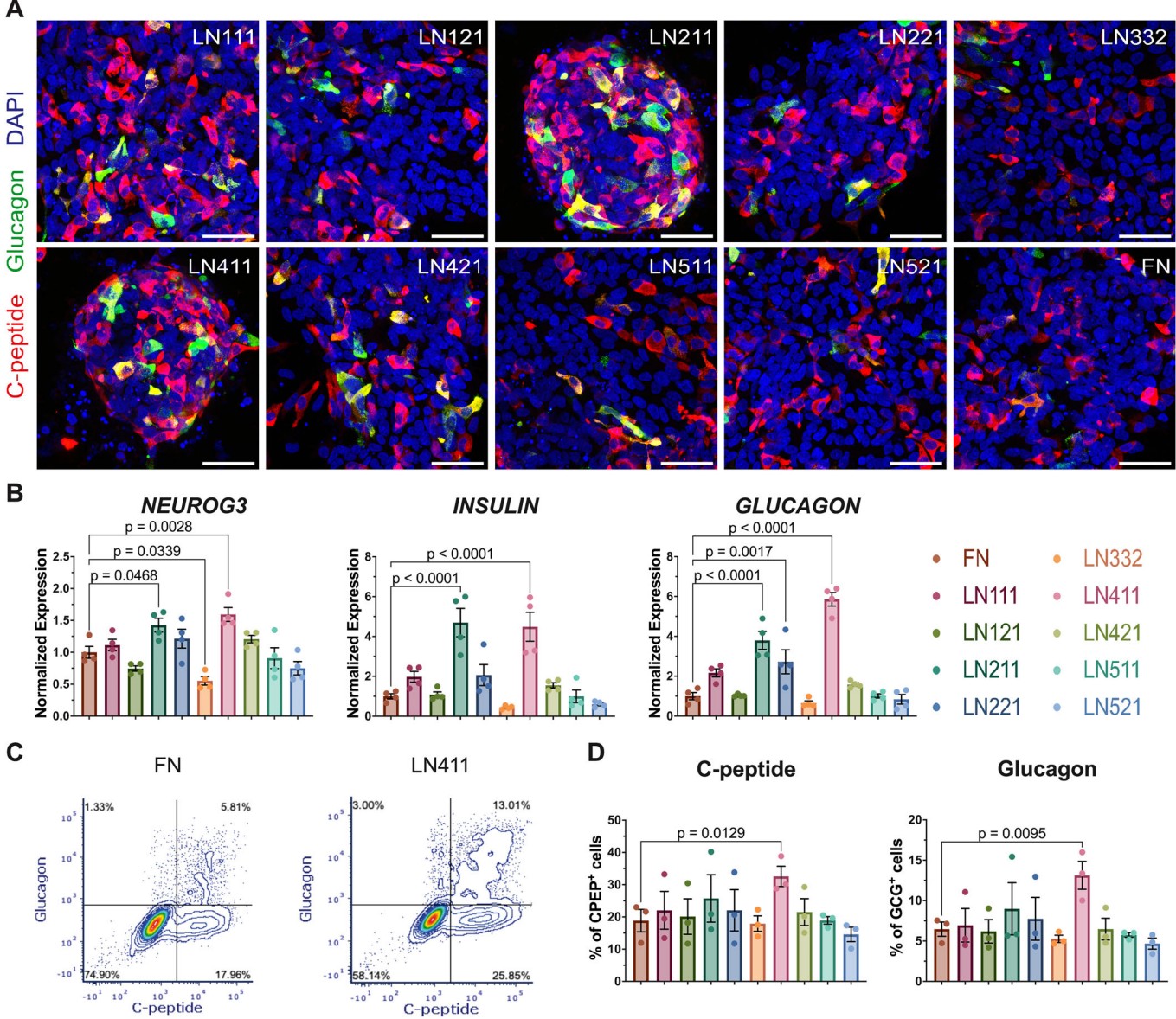

**Figure 1. LN411 specifically induces endocrine specification.**

(A) Maximum intensity projection images of HUES4-derived pancreatic progenitors reseeded on indicated ECM proteins for 9 days. Immunofluorescence with DAPI (blue), glucagon (green), and C-peptide (red). Scale bar = 50 μm. (B) qPCR analysis of endocrine marker genes (*NEUROG3*, *INSULIN*, and *GLUCAGON*) 4 days after reseeding on indicated ECM proteins. The data were analyzed by Dunnett's test with comparison to FN and are shown as mean expression ±SEM (*n* = 4, biological replicates). (C) Representative flow cytometry plots of C-peptide and Glucagon expression 9 days after reseeding on the indicated ECM proteins. (D) Quantification of total C-peptide positive cells and Glucagon positive cells in (C). The data were analyzed by Dunnett's test with comparison to FN and are shown as mean expression ± SEM (*n* = 3, biological replicates). Source data are available online for this figure.

similar phenotype of pancreatic hypoplasia coupled with expansion of the endocrine compartment occurs in the pancreata of Yap1 or Hes1 knockout mice (Mamidi et al, 2018; Jensen et al, 2000). In the case of Hes1 knockout, this phenotype has been attributed to premature specification of post-mitotic endocrine cells (Jensen et al, 2000). Together, these findings indicate that reduced expression of the laminin-binding integrins hinders the expansion of pancreatic bipotent progenitors and promotes their differentiation into the endocrine lineage over the duct lineage in a functionally redundant manner.

## Reduced cell-ECM adhesion induces endocrine specification via *PDK4*

To unravel the downstream targets responsible for the endocrinogenic impact of reduced cell-ECM adhesion, we carried out a time course bulk RNA sequencing of bipotent pancreatic progenitor cells reseeded on LN411 or fibronectin for 3, 6, or 24 h. At 3 h, only 67 genes were differentially expressed, increasing to 1065 genes at 6 h and 5780 genes at 24 h. Genes involved in endocrine specification such as *NEUROG3*, *NEUROD1*, *INSM1*, and *NKX2-2* were all only

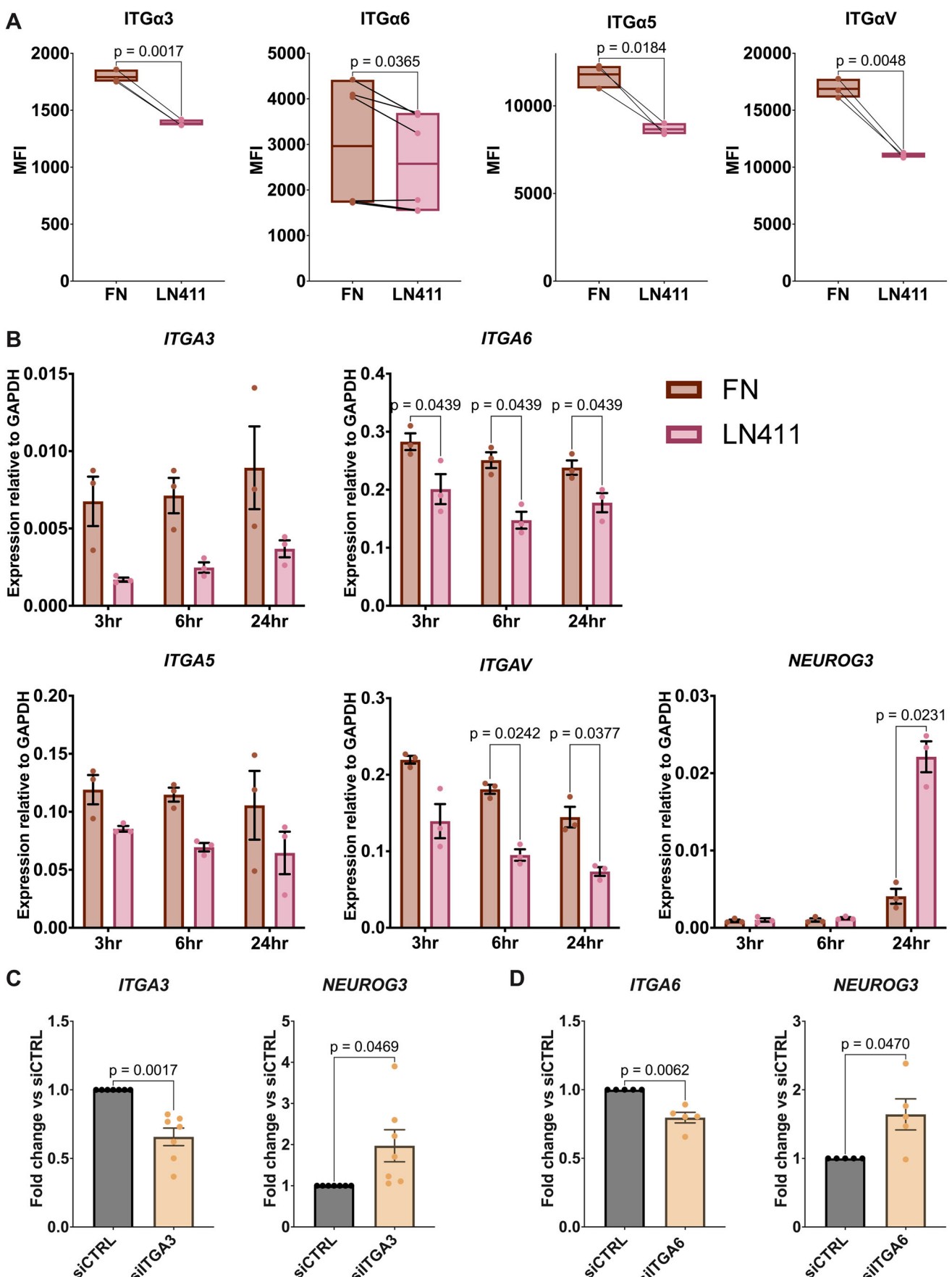

◄ **Figure 2. LN411 induces endocrine specification downstream of integrin downregulation.**

(A) Flow cytometry analysis of surface expression of indicated integrins 2 days after reseeding of NGN3-GFP-derived pancreatic progenitors on FN or LN411. Data were analyzed by two-tailed paired t-tests and are shown as boxplots with box boundaries extending to the minimum and maximum with a line at the mean (n = 6 for ITGα6, n = 3 for ITGα3, ITGα5, and ITGαV, biological replicates). (B) qPCR analysis of cell-ECM adhesion genes (ITGA6, ITGA3, ITGA5 and ITGAV) and the endocrine marker gene NEUROG3 at indicated time points after reseeding on LN411 or FN. The data were analyzed by paired t-tests with Holm–Šídák adjustment for multiple comparisons and are shown as mean expression ± SEM (n = 3, biological replicates). (C) qPCR analysis of ITGA3 and NEUROG3 2 days after transfection of pancreatic progenitors with siRNA targeting ITGA3. The data were analyzed using two-tailed paired t-tests and are shown as mean expression ± SEM (n = 7, biological replicates). (D) qPCR analysis of ITGA6 and NEUROG3 2 days after transfection of pancreatic progenitors with siRNA targeting ITGA6. The data were analyzed using two-tailed paired t-tests and are shown as mean fold change in expression ± SEM (n = 5, biological replicates). Source data are available online for this figure.

significantly upregulated after 24 h, with a minor increase in NEUROG3 expression at an earlier time point (Fig. 4A). Consistent with previous findings demonstrating YAP-mediated enhancement of endocrinogenesis upon inhibition of actin polymerization (Mamidi et al, 2018; Hogrebe et al, 2020), key YAP targets, including CYR61 (CCN1) and CTGF (CCN2), were significantly downregulated after 6 h of treatment (Fig. 4A). A similar pattern of reduced expression of YAP targets at 6 h without an increase in expression of endocrine marker genes was observed after treatment of cells with the actin-depolymerizing agent latrunculin B (Fig. EV5A).

Both LN411 exposure and latrunculin B treatment led to endocrine specification via reduced actin polymerization. LN411 acts upstream by downregulating cell-ECM adhesion, while latrunculin B directly targets actin polymerization. In both cases, YAP target genes are downregulated already at 6 h, while induction of endocrine genes is delayed. This pattern indicates the presence of previously unidentified intermediate steps in the regulation of the endocrinogenic transcriptional program. We thus hypothesized that comparing the differentially expressed genes in the latrunculin B and LN411 sequencing datasets would reveal key genes involved in the shared endocrinogenic program, while treatment-specific effects unrelated to the endocrinogenic program would be filtered out.

Of the top 100 significantly upregulated genes in each condition only nine overlapped (DDIT4, FOXA3, MET, MAN1A1, FST, FOXP1, RGS4, PDK4, and RFX6) (Fig. 4B). While FST, FOXP1, RGS4 and RFX6 have been associated with pancreatic development (Miralles et al, 1998; Spaeth et al, 2015; Serafimidis et al, 2011; Smith et al, 2010; Soyer et al, 2010), their known roles make them unlikely candidates for mediating actin depolymerization-induced endocrine specification. Specifically, FST has been shown to downregulate rather than upregulate endocrine induction (Miralles et al, 1998), while RFX6, RGS4, and FOXP1 function downstream of NEUROG3 in general endocrine development, delamination, and α-cell function, respectively (Smith et al, 2010; Serafimidis et al, 2011; Spaeth et al, 2015). In contrast, DDIT4, MAN1A1, and PDK4 have not previously been implicated in pancreas development. The decrease in PDK4 expression upon ECM attachment (Grassian et al, 2011) and its identification as a mediator of anchorage-independent growth (Tambe et al, 2019) indicate its potential significance as a relevant target for further investigation.

PDK4 encodes pyruvate dehydrogenase kinase 4, a protein serine kinase that inhibits the key metabolic enzyme pyruvate dehydrogenase, which in turn controls the conversion of pyruvate to acetyl-CoA (Zhang et al, 2014). Interestingly, PDK4 expression increased while key YAP target genes CYR61 (CCN1) and CTGF

(CCN2) decreased prior to NEUROG3 upregulation during both latrunculin B-treatment (3 h) and LN411 exposure (6 h) (Figs. 4A and EV5A).

Upregulation of PDK4 expression upon exposure to LN411 was validated using qPCR and Western blotting (Fig. EV5B–D). siRNA-based knockdown of PDK4 mRNA resulted in the downregulation of NEUROG3, thus providing genetic evidence for the functional role of PDK4 in endocrinogenesis (Fig. 4C). Consistently, treatment of bipotent pancreatic progenitor cells with the pan-PDK inhibitor dichloroacetate (DCA) reduced the expression of NEUROG3 and NEUROD1 after reseeding on LN411 (Fig. 4E) as well as without reseeding (Fig. EV5F). These effects were validated using the GFP reporter in the NGN3-GFP cell line (Figs. 4D and EV5E). Interestingly, treatment with DCA did not alter the cell morphology upon reseeding on LN411 (Fig. 4D), suggesting that cell aggregation is a consequence of upstream events, rather than a consequence of endocrinogenesis. Altogether, these data demonstrate that PDK4 is a novel effector of the cell-ECM adhesion/actin cytoskeleton-mediated regulation of endocrine specification.

## PDK4 controls endocrine specification via maintaining PDX1 expression and inducing NEUROG3 expression

To examine whether PDK4 regulates endocrine specification via YAP (Mamidi et al, 2018; Hogrebe et al, 2020), we assessed the impact of DCA treatment on YAP activity. Upon reseeding cells on LN411, DCA treatment induced higher nuclear concentrations of YAP in NKX6-1+ bipotent progenitor cells (Fig. 5A,B). This was accompanied by higher expression of the YAP target genes CYR61 and CTGF (Fig. 5C). A similar upregulation was observed in unreseeded cells following DCA treatment (Fig. EV5G). Altogether, these data indicate a key role of PDK4 in mediating reduced YAP-activity downstream of reduced cell-ECM adhesion.

To address whether PDK4 controls YAP activity via a non-canonical (Thoudam et al, 2022; Lee et al, 2022a) or metabolic mechanism, we treated pancreatic progenitors with acetate. Acetate can readily be converted into acetyl-CoA, thereby bypassing a conceivable non-canonical mechanism by PDK4. Akin to DCA treatment, acetate abrogated the expression of NEUROG3 and NEUROD1 (Figs. 6A,B and EV6A,B). However, acetate failed to increase nuclear YAP intensity in NKX6-1+ bipotent progenitor cells (Fig. 6C,D). Consistently, the expression changes of YAP target genes were marginal, with a small increase in the expression of CTGF counterbalanced by a small decrease in the expression of CYR61 (Fig. 6E). Collectively, these findings indicate that PDK4 induces endocrinogenesis through both YAP-independent metabolic and YAP-dependent non-canonical mechanisms.

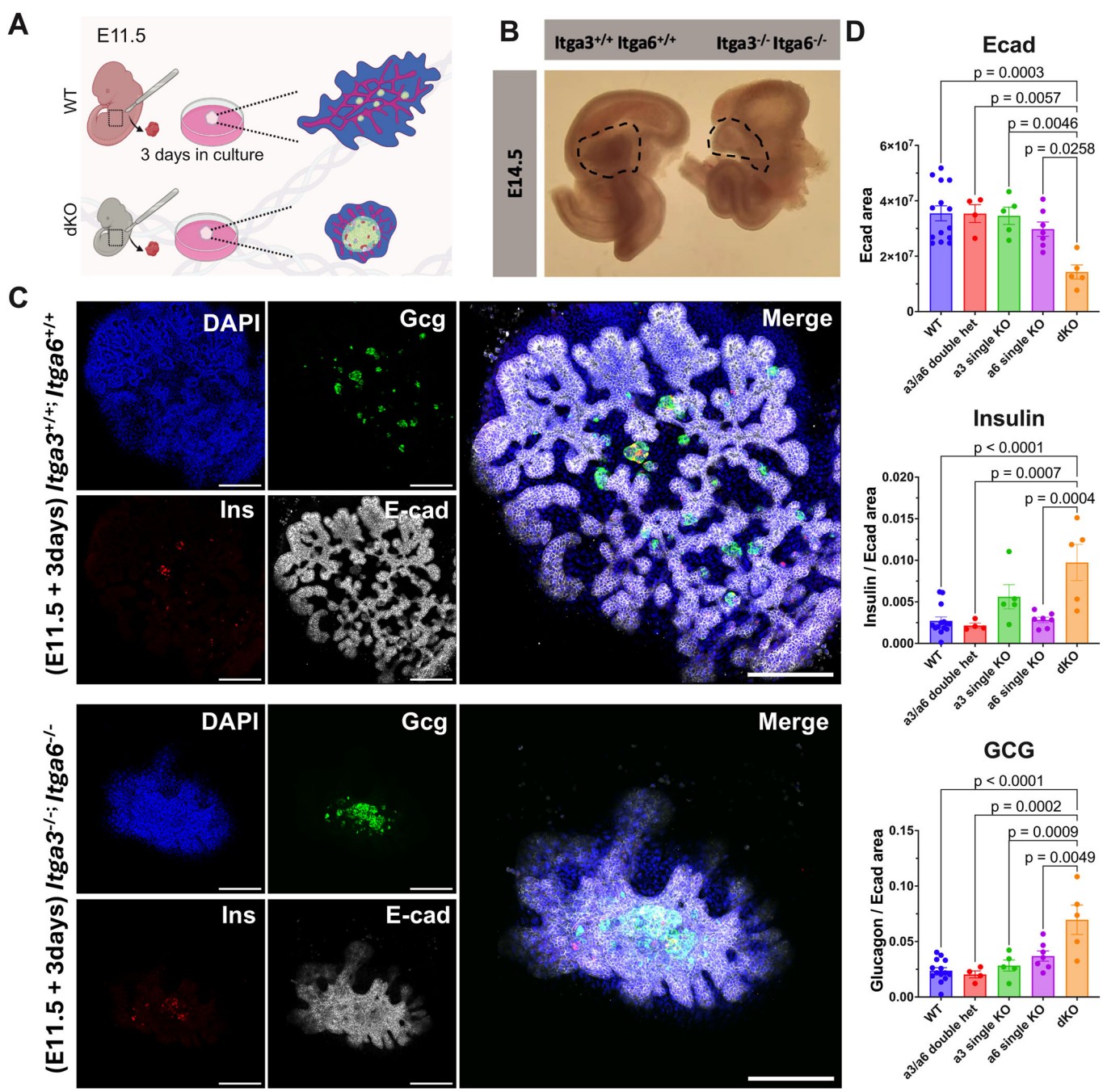

**Figure 3. ITGA3 and ITGA6 redundantly inhibit endocrine specification in vivo.**

(A) Schematic of the explant experimental setup. Due to embryonic lethality of *Itga3*$^{-/-}$; *Itga6*$^{-/-}$ mouse embryos, pancreata were explanted at E11.5. Explants were then cultured for 3 days, after which hypoplasia was apparent in double knockout explants. (B) Images showing part of the dissected gastrointestinal tract from E14.5 WT and *Itga3*$^{-/-}$; *Itga6*$^{-/-}$ mouse embryos. The double knockout exhibits hypoplasia of the pancreas (outlined). (C) Representative whole-mount staining of *Itga3*$^{+/-}$/*Itga6*$^{+/-}$, and *Itga3*$^{-/-}$/*Itga6*$^{-/-}$ pancreata explanted at E11.5 and cultured for 3 days ex vivo. Immunostaining with DAPI (blue), insulin (red), glucagon (green), and E-cadherin (E-cad, White). Scale bar = 200 µm. (D) Quantification of the staining for *Itga3*$^{+/-}$/*Itga6*$^{-/-}$ and *Itga3*$^{-/-}$/*Itga6*$^{+/-}$ pancreatic explants. The data were analyzed by Tukey's test and are shown as mean expression ± SEM ($n = 14$ for wild type, 4 for double heterozygotes, 7 for *Itga6* single knockout, and 5 for *Itga3* single knockout and *Itga3/Itga6* double knockout, biological replicates). Source data are available online for this figure.

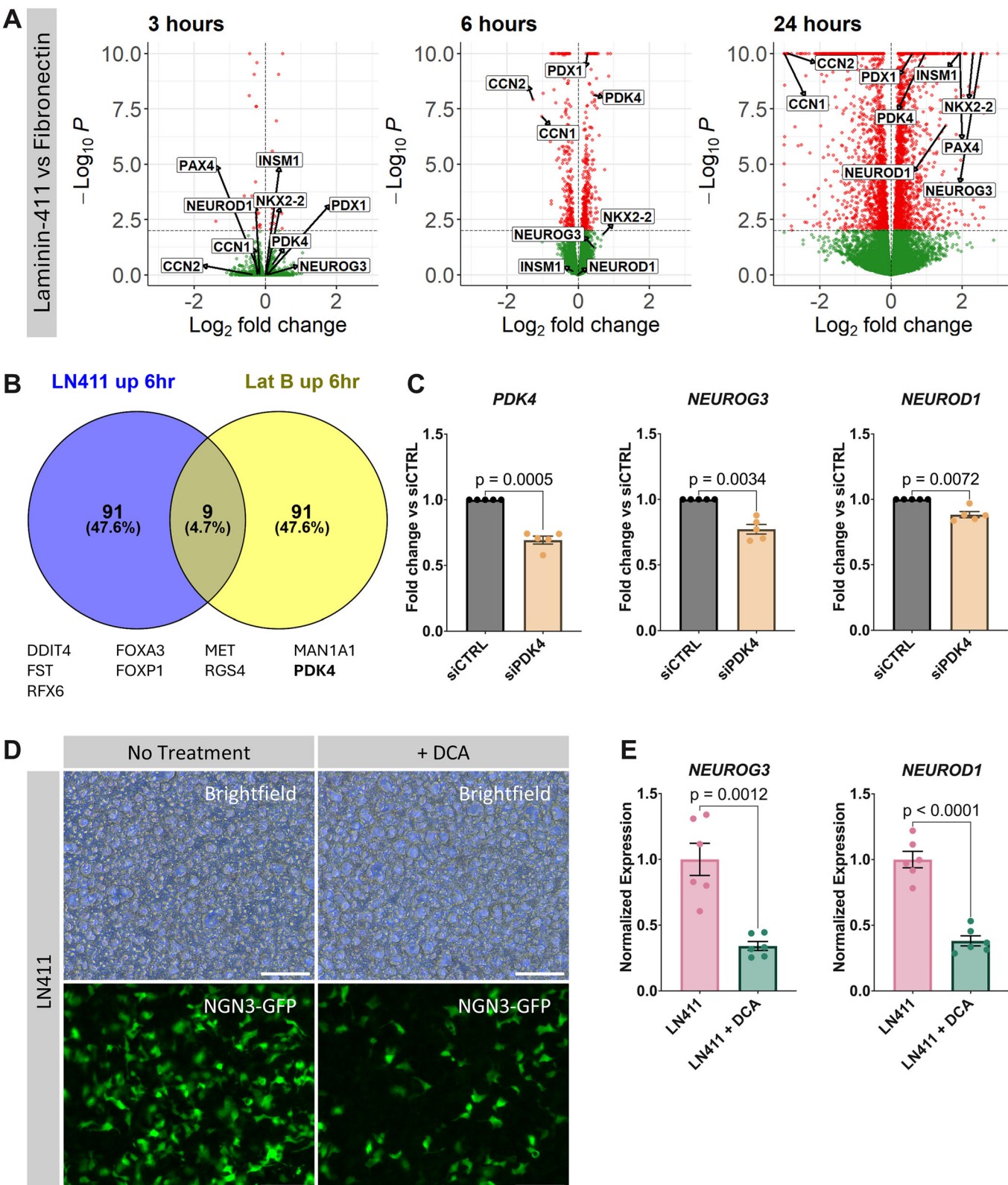

◄ **Figure 4.  LN411 induces endocrine specification via *PDK4*.**

(A) Volcano plots of bulk RNA-seq analysis of NGN3-GFP-derived pancreatic progenitors reseeded on LN411 or FN at the indicated time points. Expression of YAP target genes (*CCN1/CYR61, CCN2/CTGF*), early endocrine genes (*NEUROG3, NEUROD1, INSM1, NKX2-2*), *PDX1* and *PDK4* are indicated. Differential expression analysis was performed using DESeq2 (n = 3, biological replicates). (B) Venn diagram of the top 100 upregulated genes by significance at 6 h after reseeding on LN411 (compared to reseeding on FN) or treatment with latrunculin B (LatB) (compared to before treatment). (C) qPCR analysis of *PDK4*, and endocrine marker genes (*NEUROG3* and *NEUROD1*) 2 days after transfection with siRNA targeting *PDK4*. The data were analyzed using two-tailed paired *t*-tests and are shown as mean fold change in expression ± SEM (n = 6, biological replicates). (D) Brightfield (top panels) and epifluorescence (bottom panels) images of NGN3-GFP derived pancreatic progenitor cells 2 days after reseeding on LN411 with or without treatment with 10 mM DCA. High-density regions form a web-like structure rather than discrete circular aggregates. Scale bar = 100 μm. (E) qPCR analysis of endocrine marker genes *NEUROG3* and *NEUROD1* in pancreatic progenitors 2 days after reseeding on LN411 with or without 10 mM DCA treatment. The data were analyzed using two-tailed paired *t*-tests and are shown as mean expression ± SEM (n = 6, biological replicates). Source data are available online for this figure.

We previously observed that confinement of individual pancreatic progenitors not only reduced YAP expression and increased NEUROG3 expression, but also maintained PDX1 expression, suggesting a role of PDK4 in sustaining PDX1 expression during endocrinogenesis (Mamidi et al, 2018). Indeed, LN411 exposure increased *PDX1* expression alongside *PDK4* prior to enhanced *NEUROG3* expression (Fig. 4A). The fact that both DCA and acetate reduced *PDX1* expression suggest an involvement of PDK4 via its metabolic function (Figs. 6F and EV6D). This was corroborated by decreased intracellular levels of acetyl-CoA prior to increased *NEUROG3* expression. As expected, the LN411-mediated reduction of acetyl-CoA was blocked by concurrent treatment with acetate or DCA (Fig. EV6E).

Taken together, these findings indicate that PDK4 controls endocrine specification through two mechanisms: (1) maintenance of PDX1 expression via YAP-independent reduction of acetyl-CoA (canonical) and (2) induction of NEUROG3 via a YAP-dependent mechanism (non-canonical)(Mamidi et al, 2018) (Figs. 6G and EV7).

To assess the relevance of our findings to human in vivo development, we analyzed previously published single-cell RNA sequencing datasets. We focused on the OMIX001616 dataset of human fetal pancreata covering weeks 4–11 post conception (wpc) (Ma et al, 2023), and restricted the analysis to 8–10 wpc when a well-defined trunk population giving rise to endocrine progenitor cells was present (Zhang et al, 2025; Ma et al, 2023).

Consistent with our in vitro findings, endocrine progenitors showed significant downregulation of genes encoding integrins α3, α6, α5, and αV compared to bipotent trunk progenitors, validating reduced cell-ECM adhesion during endocrine specification (Fig. EV8A). These cells also exhibited significant upregulation of *PDX1* and a trend toward increased *PDK4* expression, alongside the previously described downregulation of YAP target genes (Fig. EV8A).

We further assessed laminin expression in the human fetal pancreas using a broader dataset including both epithelial and non-epithelial cells (Olaniru et al, 2023). *LAMA4* expression peaked at 8–10 wpc (Fig. EV8B), coinciding with the onset of endocrine induction (Ma et al, 2023; Zhang et al, 2025). Importantly, *LAMA4* expression was largely restricted to non-epithelial cells, with high expression levels in Schwann and endothelial cells and moderate expression in mesenchymal cells, making these cells the likely source of LN411 during pancreatic development (Fig. EV8B). These findings validate the gene expression changes of our proposed pathway components at the onset of endocrinogenesis during human pancreas development in vivo, with expression of *LAMA4*

at the appropriate developmental window alongside the expected decreased expression of integrins and increased expression of *PDK4* and *PDX1* in endocrine progenitor cells.

Altogether, these findings delineate a pathway linking ECM composition to endocrine fate choice via integrated mechanical and metabolic signaling pathways. First, LN411 exposure induces a broad-based downregulation of integrins α3, α6, α5 and αV, leading to reduced cell-ECM adhesion and actin depolymerization. This triggers *PDK4* upregulation, which specifies endocrine fate via non-canonical YAP inhibition and metabolically mediated *PDX1* upregulation (Fig. 6G).

## Discussion

It is well-established that cell fate decisions can be regulated by cell-ECM interactions and cellular metabolism (Mahmoud, 2023; Chi et al, 2020; Dingare et al, 2024), and that cell-ECM adhesion modulates cellular metabolism (Romani et al, 2020). However, there are no known examples of metabolic mediators mediating ECM-guided cell fate decisions. Here, we provide a new mechanism for how mechanosignaling via cell-ECM adhesion controls cell fate decisions via a metabolic regulator of glycolysis. We arrived at this conclusion by studying how endocrinogenesis in the developing pancreas is regulated by environmental influences.

Previous work demonstrated that tensile forces govern cell fate decision of bipotent pancreatic progenitors towards either an endocrine or duct fate (Mamidi et al, 2018). Here, we identify reduced cell-ECM adhesion as a major inducer of endocrinogenesis. Unexpectedly, we find that LN411 causes reduced cell-ECM adhesion through a rapid global reduction of the transcription and surface expression of multiple integrins. Thus, it is a broad lowering of integrin expression rather than targeting of specific integrin receptor-ligand signaling that triggers a mechanical cellular state conducive to endocrinogenesis. Another unknown aspect is how alterations in mechanical tension are transduced via the actin cytoskeleton into the pro-endocrinogenetic reduction in YAP signaling. Here, we identify a key metabolic kinase, PDK4, as a central player in transducing changes in cell-ECM adhesion into transcriptional regulation of endocrinogenesis. Interestingly, our results indicate that PDK4 regulates endocrinogenesis through both non-canonical and canonical mechanisms. Whereas the former represents the previously identified YAP-regulated pathway (Mamidi et al, 2018), the latter corresponds to a newly identified metabolically mediated pathway. Specifically, our findings suggest

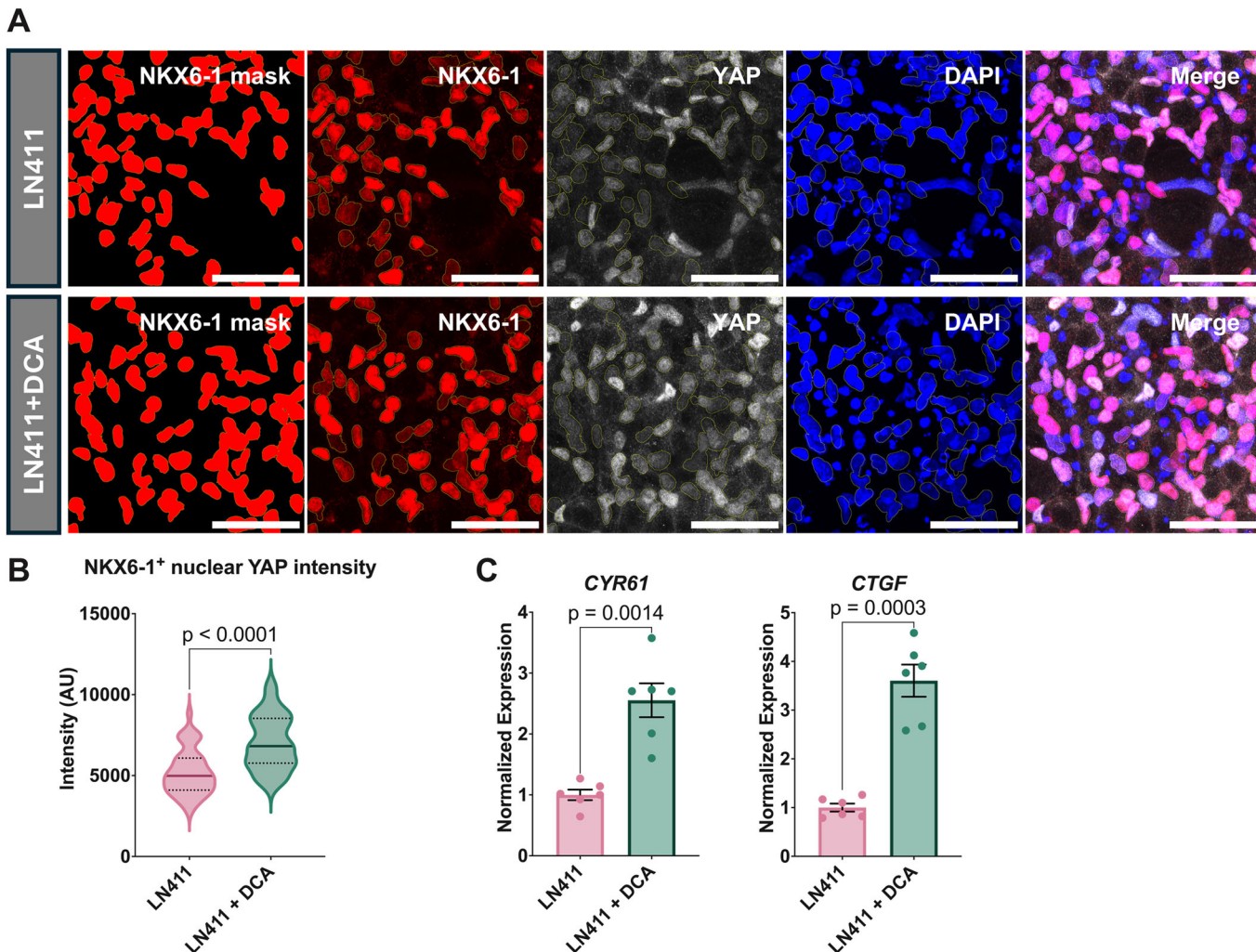

**Figure 5. The PDK4-inhibitor DCA inhibits endocrine specification via activation of YAP.**

(**A**) Representative maximum intensity projections of NGN3-GFP-derived pancreatic progenitor cells 1 day after reseeding on LN411 with or without 20 mM DCA. A mask of NKX6-1$^+$ nuclei generated using Ilastik is shown in the first panel. Immunofluorescence for NKX6-1 (red), YAP (white), and DAPI (blue). Scale bar = 50 μm. (**B**) Quantification of nuclear YAP expression in 312 NKX6-1$^+$ nuclei from four independent replicates (176 without DCA and 136 with DCA treatment). The data were analyzed using a two-tailed Welch's $t$-test. (**C**) qPCR analysis of the YAP target genes *CYR61* and *CTGF* in pancreatic progenitors 2 days after reseeding on LN411 with or without 10 mM DCA. The data were analyzed using two-tailed paired $t$-tests and are shown as mean expression ± SEM ($n = 6$, biological replicates). Source data are available online for this figure.

that PDK4 maintains PDX1 expression via reduced levels of acetyl-CoA (Fig. 6G).

The physiological relevance of our model is supported by gene expression analysis of human pancreatic development, where endocrine progenitors exhibit downregulation of *ITGA3*, *ITGA6*, *ITGA5*, and *ITGAV* and upregulation of *PDX1* and *PDK4*. The observed pattern of up- and downregulated genes in vivo perfectly matches the pattern observed during our mechanistic in vitro studies. Further, a peak in *LAMA4* expression at 8–10 wpc coincides with the timing of the onset of endocrinogenesis in vivo. Altogether, this in silico analysis supports our model that heterogeneous LN411 deposition drives local endocrine fate decisions during normal human development.

In mature human and murine islets, LN411 is deposited in the vascular basement membrane, and the main source is pericytes

(Sakhneny et al, 2021). During development, LN411 is known to be secreted by a subset of endothelial cells and by Schwann cells in the developing pancreas (Frieser et al, 1997; Olaniru et al, 2023). Single cell analysis of human pancreatic development confirms that both Schwann cells and endothelial cells express *LAMA4* at high levels during the onset of endocrinogenesis at 8–10 wpc (Olaniru et al, 2023). Schwann cells have recently been shown to co-localize with endocrine progenitor cells during human pancreas development (Olaniru et al, 2023). Whether Schwann cells play a functional role during pancreas development remains to be investigated. Existing data regarding the role of endothelial cells in endocrinogenesis is conflicting. While hypervascularization and endothelial cell depletion studies demonstrated a positive influence on endocrinogenesis by endothelial cells (Lammert et al, 2001; Pierreux et al, 2010), other studies identified mechanisms by

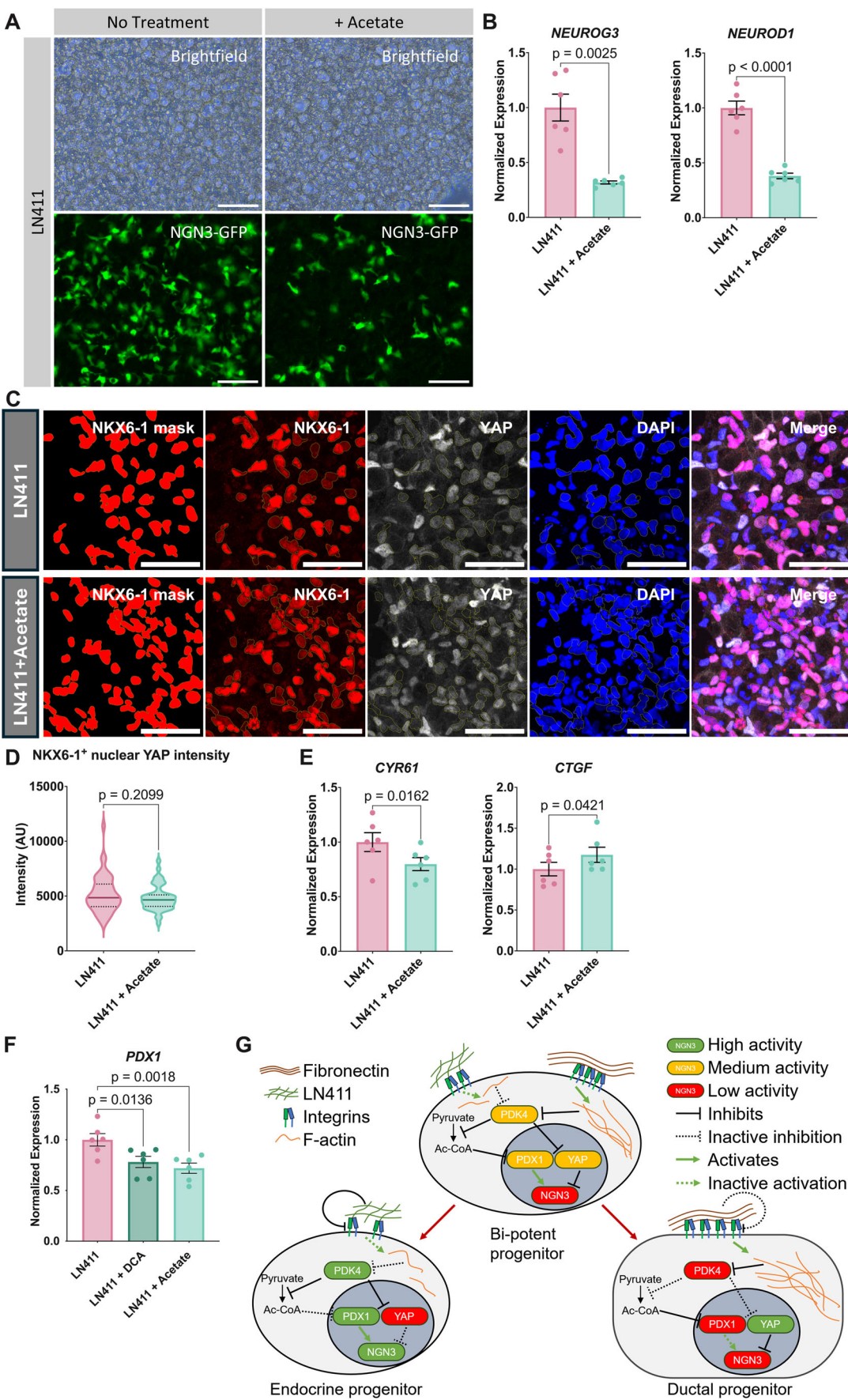

◄ **Figure 6. Acetate inhibits endocrine specification via a YAP-independent mechanism.**

(A) Brightfield (top panels) and epifluorescence (bottom panels) images of pancreatic progenitors 2 days after reseeding on LN411 with or without 10 mM sodium acetate treatment. (B) qPCR analysis of endocrine marker genes *NEUROG3* and *NEUROD1* in pancreatic progenitors 2 days after reseeding on LN411 with or without 10 mM sodium acetate treatment. The data are shown as mean expression ± SEM ($n = 6$, biological replicates). (C) Representative maximum intensity projections of NGN3-GFP derived pancreatic progenitors 1 day after reseeding on LN411 with or without 10 mM sodium acetate treatment. A mask of NKX6-1$^+$ nuclei generated using Ilastik is shown in the first panel. Immunofluorescence for NKX6-1 (red), YAP (white), and DAPI (blue). Scale bar = 50 µm. (D) Quantification of nuclear YAP expression in 253 NKX6-1$^+$ nuclei from three independent replicates (127 without sodium acetate and 126 with sodium acetate treatment). The data were analyzed using a two-tailed Welch's *t*-test. (E) qPCR analysis of the YAP target genes *CYR61* and *CTGF* in pancreatic progenitor cells 2 days after reseeding on LN411 with or without 10 mM sodium acetate treatment. The data were analyzed using two-tailed paired *t*-tests and are shown as mean expression ± SEM ($n = 6$, biological replicates). (F) qPCR analysis of *PDX1* expression in pancreatic progenitors 2 days after reseeding on LN411 with or without 10 mM acetate or DCA. The data were analyzed by Dunnett's test compared to LN411 alone and are shown as mean expression ± SEM ($n = 6$, biological replicates). (G) Model depicting the role of ECM, integrins, and PDK4 in endocrine fate choice. Exposure to LN411 induces downregulation of integrins on both the protein and RNA levels. This is turn reduces the level of actin polymerization and leads to disinhibition of *PDK4* expression. PDK4 inhibits the generation of acetyl-CoA and YAP activity. Reduced levels of acetyl-CoA promote maintenance of *PDX1* expression. The combination of YAP inhibition and high *PDX1* expression permits the expression of *NEUROG3* and endocrine fate choice. Source data are available online for this figure.

which endothelial cells inhibit endocrinogenesis (Magenheim et al, 2011; Kao et al, 2015).

Other examples of cell-ECM adhesion-mediated control of cell fate decisions via mechanical cues have been reported. For example, LN411 plays a role in the specification of retinal lineages during ocular development (Shibata et al, 2018), and of endothelial cell identity (Hall et al, 2022; Ohta et al, 2016). Although LN411's effect on endothelial cell differentiation is dependent on focal adhesion kinase, integrin-linked kinase, Notch and β-catenin signaling, the crosstalk between these pathways remains to be determined (Hall et al, 2022). Given YAP's strong connection with the Notch (Totaro et al, 2018) and Wnt signaling pathways (Azzolin et al, 2014), it is worth considering whether the LN411-PDK4-YAP axis uncovered in this study also plays a role during endothelial development.

Core metabolic enzymes often regulate cellular processes beyond their enzymatic activity—a phenomenon often referred to as "moonlighting" (Lu and Hunter, 2018). An example is PKM2, which is an isoform of the enzyme that controls the generation of pyruvate from phosphoenolpyruvate. This enzyme is part of the core glycolytic machinery but also regulates a wide range of other cellular processes through its non-canonical kinase activity, including gene expression, histone modification, mitosis, and apoptosis (Lee et al, 2022b). Here, we provide another example of the moonlighting phenomenon in which PDK4 contributes to the induction of endocrinogenesis through its non-canonical control of YAP activity. Exploring the mechanism by which PDK4 regulates YAP forms an interesting avenue for future research. One possibility is through the previously described regulation of Septin-2 by PDK4 (Thoudam et al, 2022). Septin-2 is involved in the mechanical control of YAP, specifically by being required for increased YAP activity on stiff substrates (Calvo et al, 2015).

Ample evidence supports a requirement for PDX1 in endocrinogenesis (Fujitani et al, 2006; Oliver-Krasinski et al, 2009; Zhu et al, 2016). For example, PDX1 hypomorphisms are associated with defective development of the endocrine pancreas in mice (Oliver-Krasinski et al, 2009) and human patients (Nicolino et al, 2010), suggesting that maintenance of PDX1 expression is required for endocrinogenesis in mice and man. We have previously demonstrated a strong relationship between LN-mediated cell confinement/soft environment and PDX1 expression (Mamidi et al, 2018). Based on the previously established inverse relationship between FOXA2 acetylation and PDX1 expression (Wang et al, 2013), we speculate that locally reduced cell-ECM adhesion maintains high PDX1 expression through PDK4-mediated reduction of FOXA2 acetylation through lowered acetyl-CoA synthesis from pyruvate.

Thus, the identification of PDK4 as a regulator of YAP and PDX1 via non-canonical and canonical mechanisms, respectively, highlights the need for considering the interplay between ECM-mediated mechanosignalling and metabolism in future studies of cell fate decisions of stem cells/progenitors during organogenesis.

## Methods

### Reagents and tools table

| Reagent/resource | Reference or source | Identifier or catalog number |
|---|---|---|
| **Experimental models** | | |
| C57BL/6 (*M. musculus*) | In house | NA |
| *Itga3*$^{+/-}$; *Itga6*$^{+/-}$ mice (*M. musculus*) | De Arcangelis et al, 1999 | MGI:1857792 MGI:1857793 |
| NGN3-GFP (SA121) ES cells (*H. sapiens*) | Löf-Öhlin et al, 2017 | NA |
| HUES4 ES cells (*H. sapiens*) | Harvard University | BioSamples: SAMEA114214019 |
| **Antibodies** | | |
| Anti-insulin (1:400) | DAKO | #A0564 RRID:AB_10013624 |
| Anti-E-cadherin (1:400) | Takara | #M108 RRID:AB_2895157 |
| Anti-glucagon (1:1000) | Sigma | #G2654 RRID:AB_259852 |
| Anti-ITGA3 (1:100) | R&D | #AF2787 RRID:AB_2129595 |
| Anti-ITGA6 (1:200) | Millipore | #MAB1378 RRID:AB_2128317 |
| Anti-ITGA5 (1:100) | Abcam | #ab-150361 RRID:AB_2631309 |
| Anti-ITGAV (1:100) | Abcam | #ab-63490 RRID:AB_1140041 |
| Anti-COLIV (1:100) | Merck | #AB769 RRID:AB_92262 |
| Anti-LAMA1 (1:100) | T. Sasaki, Oita University | #1057+ RRID:NA |

| Reagent/resource | Reference or source | Identifier or catalog number |
|---|---|---|
| Anti-LAMA2 (1:100) | T. Sasaki, Oita University | #1124+ RRID:NA |
| Anti-LAMA4 (1:100) | T. Sasaki, Oita University | #1100+ RRID:NA |
| Anti-LAMA5 (1:100) | T. Sasaki, Oita University | #1121+ RRID:NA |
| Anti-Pdx1 (1:200) | Abcam | #ab-47308 RRID:AB_777178 |
| Anti-NKX6-1 (1:200) | DSHB | #F55A12 RRID:AB_532379 |
| Anti-YAP (1:100/ 1:200/1:1000) | Cell Signaling | #4912 RRID:AB_2218911 |
| Anti-glucagon (1:1000) | Sigma | #G2654 RRID:AB_259852 |
| Anti-C-peptide (1:200) | DSHB | #GN-ID4-S RRID:AB_2255626 |
| Anti-proglucagon (1:250) | Cell Signaling | #8233S RRID:AB_10859908 |
| PE anti-CD49f (ITGA6) (5 µl/1 million cell) | BD Biosciences | #555736 RRID:AB_396079 |
| APC anti-CD49e (ITGA5) (3 µl/1 million cell) | Biolegend | #328002 RRID:AB_893363 |
| PE anti-CD49c (ITGA3) (5 µl/1 million cell) | Biolegend | #343803 RRID:AB_1731941 |
| APC anti-CD51 (ITGAV) (5 µl/1 million cell) | Biolegend | #327913 RRID:AB_2876633 |
| APC IgG2a, κ (5 µl/1 million cell) | Biolegend | #400221 RRID:AB_2891178 |
| PE IgG1, κ (5 µl/1 million cell or 10 µl/10 million cell) | BD Biosciences | #555749 RRID:AB_396091 |
| APC IgG2b, κ (3 µl/1 million cell) | Biolegend | #400311 RRID:AB_2894969 |
| PE IgG2a, κ (1:200) | BD Biosciences | #553927 RRID:AB_395142 |
| PE anti-GP2 (10 µl/ 10 million cells) | Nordic Biosite | #D277-5 RRID:AB_11160953 |
| Anti-PDK4 (1:5000) | Proteintech | #12949-1-AP RRID:AB_2161499 |
| Anti-vinculin (1:1000) | Sigma | #V9264 RRID:AB_10603627 |
| Goat anti-rabbit IgG (H + L) secondary antibody, HRP | Thermo Fisher | #31460 RRID:AB_228341 |
| Goat anti-mouse IgG (H + L) secondary antibody, HRP | Thermo Fisher | #31430 RRID:AB_228307 |
| **Oligonucleotides and other sequence-based reagents** | | |
| *GAPDH* TaqMan probe | Thermo Fisher | Hs02758991_g1 |
| *ITGA3* TaqMan probe | Thermo Fisher | Hs01076879_m1 |
| *ITGA5* TaqMan probe | Thermo Fisher | Hs01547673_m1 |
| *ITGA6* TaqMan probe | Thermo Fisher | Hs01041011_m1 |
| *ITGAV* TaqMan probe | Thermo Fisher | Hs00233808_m1 |
| *PDX1* TaqMan probe | Thermo Fisher | Hs00236830_m1 |
| *NEUROG3* TaqMan probe | Thermo Fisher | Hs01875204_s1 |
| *NEUROD1* TaqMan probe | Thermo Fisher | Hs00159598_m1 |
| *INS* TaqMan probe | Thermo Fisher | Hs02741908_m1 |
| *GCG* TaqMan probe | Thermo Fisher | Hs01031536_m1 |
| *CYR61* TaqMan probe | Thermo Fisher | Hs00155479_m1 |
| *CTGF* TaqMan probe | Thermo Fisher | Hs01026927_g1 |
| *RPL37A* TaqMan probe | Thermo Fisher | Hs01102345_m1 |
| *PDK4* TaqMan probe | Thermo Fisher | Hs01037712_m1 |
| *PDK4* siRNA | Thermo Fisher | s10262 |
| *ITGA3* siRNA | Thermo Fisher | s7541 |
| *ITGA6* siRNA | Thermo Fisher | s7494 |
| Negative control siRNA | Thermo Fisher | #4392420 |
| **Chemicals, enzymes and other reagents** | | |
| CHIR99021 | Axon Medchem | #1386 |
| Activin A | PeproTech | #120-14 |
| FGF2 | PeproTech | #100-18B |
| Retinoic acid | Sigma Aldrich | #R2625 |
| TBP | Calbiochem | #565740 |
| ALK5ill | Santa Cruz | #sc-221234A |
| Noggin | PeproTech | #120-10 C |
| Nicotinamide | Sigma Aldrich | #481907 |
| Forskolin | Sigma Aldrich | #F6886 |
| Latrunculin B | Sigma Aldrich | #L5288 |
| Bovine serum albumin (BSA) | Sigma Aldrich | #B4287 |
| Y-27632 | Merck | #688000 |
| Biolaminin 521 LN (LN521) | BioLamina | #LN521-05 |
| Biolaminin 411 LN (LN411) | BioLamina | #LN411-02 |
| B27 supplement minus Insulin | Thermo Fisher | #A1895601 |
| B27 supplement | Thermo Fisher | #17504001 |
| NutriStem hPSC XF, xeno-free medium | Sartorius | #05-100-1 A |
| StemPro™ Accutase™ | Thermo Fisher | #A1110501 |
| Fibronectin (cell experiments) | Sigma | #F0895 |
| Fibronectin (explant experiments) | Life Technologies | #33010-018 |
| M-199 Media + phenol red | Thermo Fisher | #31150022 |
| Lipofectamine 2000 | Invitrogen | #11668019 |
| Lipofectamine RNAiMax | Invitrogen | #13778100 |

| Reagent/resource | Reference or source | Identifier or catalog number |
|---|---|---|
| SuperScript III | Thermo Fisher | #18080085 |
| RIPA buffer | Thermo Fisher | #8990 |
| Laemmli reducing SDS buffer | Thermo Fisher | #J60015.AC |
| Protease inhibitor | Thermo Fisher | #A32953 |
| **Software** | | |
| Adobe Illustrator 2023 | Adobe | |
| Fiji 2.0/ImageJ | http://imagej.nih.gov/ij NIH Image | |
| Venny 2.1.0 | https://bioinfogp.cnb.csic.es/tools/venny/ Oliveros, 2007 | |
| ILASTIK v1.4.0.post1 | https://ilastik.org Berg et al, 2019 | |
| GraphPad Prism 10 | https://www.graphpad.com GraphPad | |
| FCS express 7 | https://denovosoftware.com De Novo Software | |
| RStudio | https://posit.co/downloads/ RStudio | |
| R package DESeq2 v1.34.0 | https://bioconductor.org/packages/release/bioc/html/DESeq2.html Love et al, 2014 | |
| R package MAST v1.30.0 | Finak et al, 2015 | |
| R package Seurat v5.3.0 | https://cran.r-project.org/web/packages/Seurat/index.html Hao et al, 2023 | |
| STAR v2.7.2 d | https://github.com/alexdobin/STAR/releases Dobin et al, 2013 | |
| FastQC v0.11.9 | http://www.bioinformatics.babraham.ac.uk/projects/fastqc Babraham Bioinformatics group | |
| multiqc v1.7 | https://multiqc.info Ewels et al, 2016 | |
| 3DSuite | https://imagej.net/plugins/3d-imagej-suite/ Ollion et al, 2013 | |
| Single-cell transcriptomic and spatial map of the human fetal pancreas Shiny App | https://www.humanpancreasdevelopment.org/ (Olaniru et al, 2023) | |
| **Other** | | |
| Ibidi μ-Slide 8 well^high slides | Ibidi | #80806 |
| Ibidi 12-well slides for imaging | Ibidi | #80801 |
| Pierce™ BCA Protein Assay Kit | Thermo Fisher | #23227 |
| RNeasy Micro Kit | Qiagen | #74004 |
| RNeasy Mini Kit | Qiagen | #74106 |
| iScript cDNA Synthesis Kit | BIO-RAD | #1708891 |
| Applied Biosystems TaqMan Fast Universal PCR Master Mix (2X), no AmpErase UNG | Thermo Fisher | #4352042 |

| Reagent/resource | Reference or source | Identifier or catalog number |
|---|---|---|
| NEBNext Ultra II RNA Library Prep Kit for Illumina | Bionordika | #E7770S |
| NEBNext Poly(A) mRNA Magnetic Isolation Module | Bionordika | #E7490S |
| LIVE/DEAD™ Fixable Violet Dead Cell Stain Kit | Thermo Fisher | #L34964 |
| A647 Phalloidin | Thermo Fisher | #A22287 |
| 8–16% polyacrylamide gels | Bio-Rad | #4561105 |
| Chemiluminescent HRP substrate kit | Thermo Fisher | #34579 |
| Acetyl-CoA ELISA kit | Biomol | #G-AEES00006.96 |
| Zeiss LSM 780 | Zeiss | |
| Leica Microsystems DM-IL-LED microscope | Leica Microsystems | |
| QuantStudio 7 Flex | Applied Biosystems | |
| StepOne plus real-time qPCR system | Applied Biosystems | |
| Illumina NextSeq 2000 | Illumina | |
| Illumina NextSeq 500 | Illumina | |
| BD LSRFortessa | BD Biosciences | |
| Sony SH800 | Sony | |
| BD FACS Aria III | BD Biosciences | |
| Varioskant™ LUX Multimode Microplate Reader | Thermo Fisher | |

## Methods and protocols

### Stem cell culture and directed differentiation

HUES4 and NGN3-GFP-SA121 hESC cell lines were maintained on laminin-521 with daily changes of NutriStem medium at 37 °C and 5% $CO_2$. Cells were passaged two to three times weekly using Accutase at 80–90% confluency. The medium was supplemented with 10 μM ROCK inhibitor on the first day after passaging. Cells were seeded on laminin-521 for pancreatic differentiation at 80 K cells/cm². At 100% confluency, cells were induced to differentiate into pancreatic progenitor cells following a previously published protocol (Ameri et al, 2017). Treatments and reseeding were performed on day 19. Endocrine progenitors were generated by the addition of 0.5 μM latrunculin B from days 19–21 and culture until day 28. For reseeding, pancreatic progenitor cells were dissociated with accutase for 10 min at 37 °C and plated in Ibidi 8 or 12-well slides for imaging, or in 24-well plates for other analyses. In each case, the slides were precoated overnight at 4 °C with either 0.1 μg/μl fibronectin or about 0.75 μg/cm² of indicated laminins.

For knockdown experiments, HUES4 cells (*ITGA3*, *ITGA6*) or NGN3-GFP cells (*PDK4*) were reseeded as described above. For knockdown of integrins, reseeded cells were plated on LN521-

coated 24-well plates with prepared siRNA-lipofectamine 2000 complexes. For knockdown of *PDK4*, cells were mixed with prepared siRNA-lipofectamine RNAiMax complexes and plated on LN411-coated 24-well plates. Transfected cells were harvested after 48 h for gene expression analysis.

## Mouse work

Animal work was conducted at the Department of Experimental Medicine at Copenhagen University following the ethical regulations approved by the Danish animal inspectorate (Miljø- og fødevareministeriet—Dyrforsøgstilsynet, approval #2016-15-0201-01114) or at the animal facility at the IGBMC institute at Strasbourg University campus of Illkirch.

*Itga3; Itga6* double knockout mice (De Arcangelis et al, 1999) were maintained on a heterozygous background due to embryonic lethality. $Itga3^{+/-}; Itga6^{+/-}$ mice were intercrossed to generate the double mutants. Littermates lacking one integrin allele were considered wild type, $Itga3^{+/-}; Itga6^{-/-}$ embryos were considered *Itga6* single knockout, and $Itga3^{-/-}; Itga6^{+/-}$ embryos were considered *Itga3* single knockout.

Mice were kept on a C57BL/6 background. The day of vaginal plug was recorded as day 0.5 of gestation. Both male and female embryos were included in the experiments.

## Ex vivo explant culture

The dorsal pancreata of E11.5 mouse embryos were microdissected and cultured (Percival and Slack, 1999) on ibidi μ-Slide 8 well[high] slides precoated for 45 min with 0.1 μg/μl fibronectin. The explant culture media was composed of M-199 media + phenol red supplemented with 10% fetal bovine serum (FBS), 1% penicillin/streptomycin, and 0.5% Fungizone. The media was changed every other day. Explants were cultured at 37 °C for 3–4 days.

## qPCR

RNA was extracted using either the RNeasy Mini kit or RNeasy Micro kit, depending on the expected RNA yield. In both cases, on-column DNAse treatment was performed. Reverse transcription was performed using either SuperScript III using 2.5 mM random hexamer and 2.5 mM oligo(dT) or using the iScript cDNA synthesis kit, in both cases according to the manufacturer's instructions.

Real-time quantitative PCR was conducted as either duplicates on a StepOne Plus Real-time qPCR system or as triplicates on a QuantStudio 7 Flex system using TaqMan Fast Universal PCR Master Mix. Relative gene expression was determined using either GAPDH or RPL37A.

## RNA sequencing

RNA was extracted for qPCR analysis. RNA integrity was assessed using an Agilent Fragment Analyzer. Sequencing libraries were carried out according to the manufacturer's instructions using NEBNext Ultra II RNA Library Prep Kit for Illumina, along with the NEBNext Poly(A) mRNA Magnetic Isolation Module. Sequencing was run on an Illumina NextSeq 500 (latrunculin B-treated samples) or Illumina NextSeq 2000 (FN/LN411-exposed samples).

Fastq files were generated using bcl2fastq v2.20.0 and aligned to the hg38/ GRCh38.p13 genome using STAR (Dobin and Gingeras, 2015). Transcript expression levels were estimated with the --quantMode GeneCounts option and gencode.v32 annotations. FastQC was used for QC metrics, and multiqc (Ewels et al, 2016) for reporting. Data analysis was then performed with R/Bioconductor (R Core Team, 2021; Gentleman et al, 2004). Normalization and gene differential expression analysis were performed with DESeq2 (Love et al, 2014).

Sequencing datasets have been deposited to GEO.

## Flow cytometry

Cells were dissociated using Accutase for 5–10 min at 37 °C and quenched with medium. Cells were then stained for 30 min on ice with LIVE/DEAD fixable violet. Cells were then either assayed directly for expression of NGN3-GFP, incubated with conjugated antibodies against surface markers or matched isotype controls for 1 h on ice prior to assay of NGN3-GFP and surface marker expression, or incubated for 20 min in 4% paraformaldehyde on ice for fixation. Fixed cells were washed with 1% BSA in PBS, and then incubated for 1 h on ice in permeabilization/blocking buffer [0.1% Triton X-100, 5% normal donkey serum (NDS)]. The cells were then incubated overnight at 4 °C with primary antibodies or isotype controls in blocking buffer. Cells were then washed and incubated for 1 h at room temperature with secondary antibodies. All secondary antibodies used were raised in donkey and purchased from Jackson Immunoresearch. Finally, cells were washed and resuspended in 1% BSA in PBS. All assays were performed on a BD LSRFortessa. Data was analyzed using FCS Express 7.

## Fluorescently activated cell sorting

Pancreatic progenitor cells were differentiated until the late pancreatic progenitor stage (day 19), dissociated with Accutase for 5–10 min at 37 °C, and quenched with pancreatic progenitor stage medium. Cells were incubated with 5% (NDS) in PBS$^{-/-}$ for 45 min on ice for blocking. Next, cells were pelleted, resuspended in FACS buffer (0.1% bovine serum albumin (BSA), 2 mM EDTA, 10 μM ROCK inhibitor Y-27632, and 50U Penicillin/Streptomycin in PBS$^{-/-}$) and incubated with mouse anti-GP2-PE for 1 h on ice. DAPI was used to stain dead cells. Cells were sorted on Sony SH800 or BD FACS Aria III cell sorters, with gating for live GP2[high] cells performed as described previously (Ameri et al, 2017). Cells were sorted into FACS buffer, pelleted, resuspended in pancreatic progenitor medium supplemented with 10 μM Y-27632, and seeded at 50,000 cells/cm². Cells were cultured for 48 h before fixation or harvesting for subsequent analyses.

For analysis of sorted NGN3-GFP$^+$ cells, cells were sorted into NGN3-GFP$^-$/GP2[high] and NGN3-GFP$^+$ populations and were pelleted and harvested immediately.

## Immunostaining and microscopy

For immunohistochemistry, mouse pancreata were dissected at E15.5, fixed, and sectioned as previously discussed (Kesavan et al, 2009). Tissues were sectioned at 20 μm using a Leica cryostat. Sections were dried for 15 min at 37 °C, washed in PBS$^{+/+}$ and incubated with blocking buffer (PBS$^{+/+}$ + 0.1% Triton X-100 + 5% NDS) for 3–4 h. Sections were then incubated with primary

antibodies in blocking buffer overnight at 4 °C in a humidified chamber. After washing, sections were incubated with secondary antibodies in blocking buffer for 1 h at RT. Sections were rewashed and mounted with fluorescent mounting medium. For staining of whole explants, a fixation of 30 min in 4% PFA at room temperature was followed by overnight blocking at 4 °C, three nights of incubation with primary antibodies at 4 °C, and overnight incubation with secondary antibodies at 4 °C.

In vitro generated cells were permeabilized at room temperature for 15 min in permeabilization buffer (PBS$^{+/+}$ + 0.5% Triton X-100), followed by blocking for 1 h with blocking buffer and overnight incubation with primary antibodies in blocking buffer. Samples were then washed three times in washing buffer (PBS$^{+/+}$ + 0.1% Triton X-100) and incubated for 1 h at room temperature with secondary antibodies and, in indicated samples, phalloidin. Cells were then washed three times in washing buffer, with DAPI added in the penultimate washing step.

Confocal images were acquired using a Zeiss laser scanning microscope (LSM 780) using Plan-Apochromat 20X/0.8 M27 and Plan-Apochromat 40X/1.20 water immersion objectives or Leica SP8 using HC PL APO CS2 20X/0.75 IMM objective. Tilescan images of the whole pancreatic explant area were acquired using a 20X/0.8 M27 objective. Images were stitched using Zen software and quantified for insulin and glucagon using Fiji. Insulin and glucagon ratios were measured by normalizing the integrated density, which considers the area and the mean gray value, of insulin and glucagon against the integrated density of E-cadherin. Brightfield and epifluorescence images were acquired using a Leica Microsystems DM-IL-LED microscope. For images presented in Figs. 4D, 6A, EV5E, and EV6A, the subtract background function in FIJI was applied with a rolling ball radius of 1000 pixels.

Time-lapse imaging was done on Zeiss LSM 780 using Plan-Apochromat 20X/0.8 M27. HUES4 cells were reseeded on LN411 or FN and incubated in a pre-heated humidified chamber (37 °C, 5% CO$_2$). Imaging was started 1 h after reseeding. Z-stacks of differential-interference contrast (DIC) were acquired for each marked position every 15 min for 24 h.

## Quantification of nuclear YAP expression

NKX6-1 nuclei were segmented using a random forest classifier trained using sparse annotations in Ilastik (Berg et al, 2019). All possible features were enabled. After obtaining an image mask, YAP intensity in the segmented nuclei was quantified three-dimensionally using 3D suite in FIJI (Ollion et al, 2013).

## Western blotting

Cells were washed twice, collected, and lysed on ice for 30 min using RIPA buffer. Lysate was then sonicated for 1 min at 5 s intervals and centrifuged at 10,000 × $g$ at 4 °C, after which the supernatant was transferred into new tubes. Concentration was determined by bicinchoninic acid, and protein samples were boiled for 5 min in Laemmli reducing SDS buffer. Samples were separated by SDS-PAGE on 8–16% polyacrylamide gels and transferred to PVDF membranes. Membranes were blocked for 1 h at room temperature in a blocking buffer consisting of 3% non-fat milk in TBS and incubated overnight with primary antibodies in the same blocking buffer. After washing,

membranes were incubated for 1 h at room temperature with horse-radish peroxidase-conjugated secondary antibodies. Bands were detected by a chemiluminescent HRP substrate kit. Bands were then quantified by densitometry using FIJI.

## Acetyl-CoA measurement by ELISA

Cells were washed once with PBS and lysed on ice for 10 min in lysis buffer containing 0.5% NP-40, 50 mM Tris-HCl (pH 7.4), 150 mM NaCl, 1 mM EDTA and 1 × Protease inhibitor. Lysates were centrifuged at 10,000 × $g$ for 10 min at 4 °C to remove debris. The supernatant was collected and immediately used or stored at −80 °C. Acetyl-CoA levels were quantified using the acetyl-CoA ELISA kit following the manufacturer's protocol. Samples and standards were measured in duplicates, and absorbance was recorded at 450 nm using the Varioskant™ LUX Multimode Microplate Reader. Final concentrations were normalized to total protein content determined by BCA assay.

## Single-cell RNA-seq data analysis

The OMIX001616 dataset (Ma et al, 2023), containing human fetal pancreata from weeks 4–11 post conception, was downloaded from the CNCB website. Analysis was restricted to 8–10 wpc and previously annotated "Trunk" and "EP" (endocrine progenitor) populations. Data processing was performed using R with Seurat v5.3.0. Statistical analysis was performed using MAST v1.30.0, with $p$-values adjusted using the Benjamini–Hochberg method to account for multiple comparisons. Expression of laminin α isoforms was assessed using the Shiny app at humanpancreasdevelopment.org (Olaniru et al, 2023).

## Statistical analysis

Statistical analyses were performed using GraphPad Prism version 10. Two-sample comparisons were performed using two-tailed Student's $t$-tests. When possible, pairing according to the differentiation batch was performed. In other cases, Welch's $t$-test was employed. Experiments with more than two samples were analyzed by one-way repeated measures ANOVA with Geisser-Greenhouse correction, followed by Dunnett's multiple comparison test. For ELISA analysis of acetyl-CoA levels, two-way repeated measures ANOVA was performed with Geisser-Greenhouse correction, followed by Dunnett's test. Significance was defined as $p < 0.05$. Exact $p$ values are shown on graphs unless $p < 0.0001$.

# Data availability

The RNA sequencing data generated in this study has been deposited in the Gene Expression Omnibus (GEO) under accession codes GSE275775 (latrunculin B- and DMSO-treated samples) and GSE279067 (FN/LN411-exposed samples).

The source data of this paper are collected in the following database record: biostudies:S-SCDT-10_1038-S44319-025-00610-6.

## Peer review information

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

## Acknowledgements

We thank Anna Øster, Jette Larsen, and Diana Bach for her excellent technical support, Anna Månsson for helping with animal experiments, and Ivan Kulik for input and assistance with immunofluorescence experiments. We would further like to thank Gelo Dela Cruz and Paul van Dieken of the DanStem Flow Cytometry platform, Anup Shrestha and Jutta Bulkescher of the DanStem Imaging Platform, and Helen Neil, Heike Wollmann, Adrija Kalvisa, and Magali

Michaut of the DanStem Genomics Platform. All the work is supported by the European Union's Horizon 2020 research and innovation program (ISLET, no. 874839), the Novo Nordisk Foundation Center for Stem Cell Biology (DanStem) at the University of Copenhagen (NNF grant, NNF17CC0027852) and the Helmholtz Zentrum München.

## Author contributions

**Christine Ebeid**: Conceptualization; Formal analysis; Investigation; Visualization; Methodology. **Adam Rump**: Conceptualization; Data curation; Formal analysis; Investigation; Visualization; Methodology; Writing—original draft; Writing—review and editing. **Chenglei Tian**: Conceptualization; Investigation; Visualization; Methodology; Writing—review and editing. **Anant Mamidi**: Conceptualization; Supervision; Investigation. **Adèle De Arcangelis**: Resources; Writing—review and editing. **Gérard Gradwohl**: Resources; Writing—review and editing. **Henrik Semb**: Conceptualization; Supervision; Funding acquisition; Methodology; Project administration; Writing—review and editing.

Source data underlying figure panels in this paper may have individual authorship assigned. Where available, figure panel/source data authorship is listed in the following database record: biostudies:S-SCDT-10_1038-S44319-025-00610-6.

## Funding

## Disclosure and competing interests statement

The authors declare no competing interests.

# Expanded View Figures

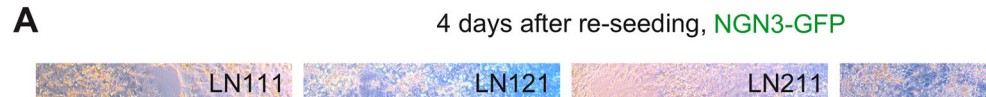

**A**                          4 days after re-seeding, NGN3-GFP

(brightfield and epifluorescence panels: LN111, LN121, LN211, LN221, LN332, LN411, LN421, LN511, LN521, FN)

**B**     FN          LN411        **C**    NGN3-GFP

(flow cytometry plots: FN NGN3-GFP 25.89%; LN411 NGN3-GFP 46.48%)

Bar chart legend:
- FN
- LN111
- LN121
- LN211
- LN221
- LN332
- LN411
- LN421
- LN511
- LN521

p = 0.0167

**Figure EV1.   LN411 specifically induces endocrine specification in the NGN3-GFP cell line.**

(A) Representative brightfield (first and third row of panels) and epifluorescence (second and fourth row of panels) images of NGN3-GFP-derived pancreatic progenitors 4 days after reseeding on the indicated ECM proteins. Scale bar = 200 µm. (B) Representative flow cytometry plots of NGN3-GFP reporter expression 4 days after reseeding. (C) Quantification of NGN3-GFP reporter expression in (B). The data were analyzed by Dunnett's test with comparison to FN and are shown as mean expression ± SEM (n = 4, biological replicates).

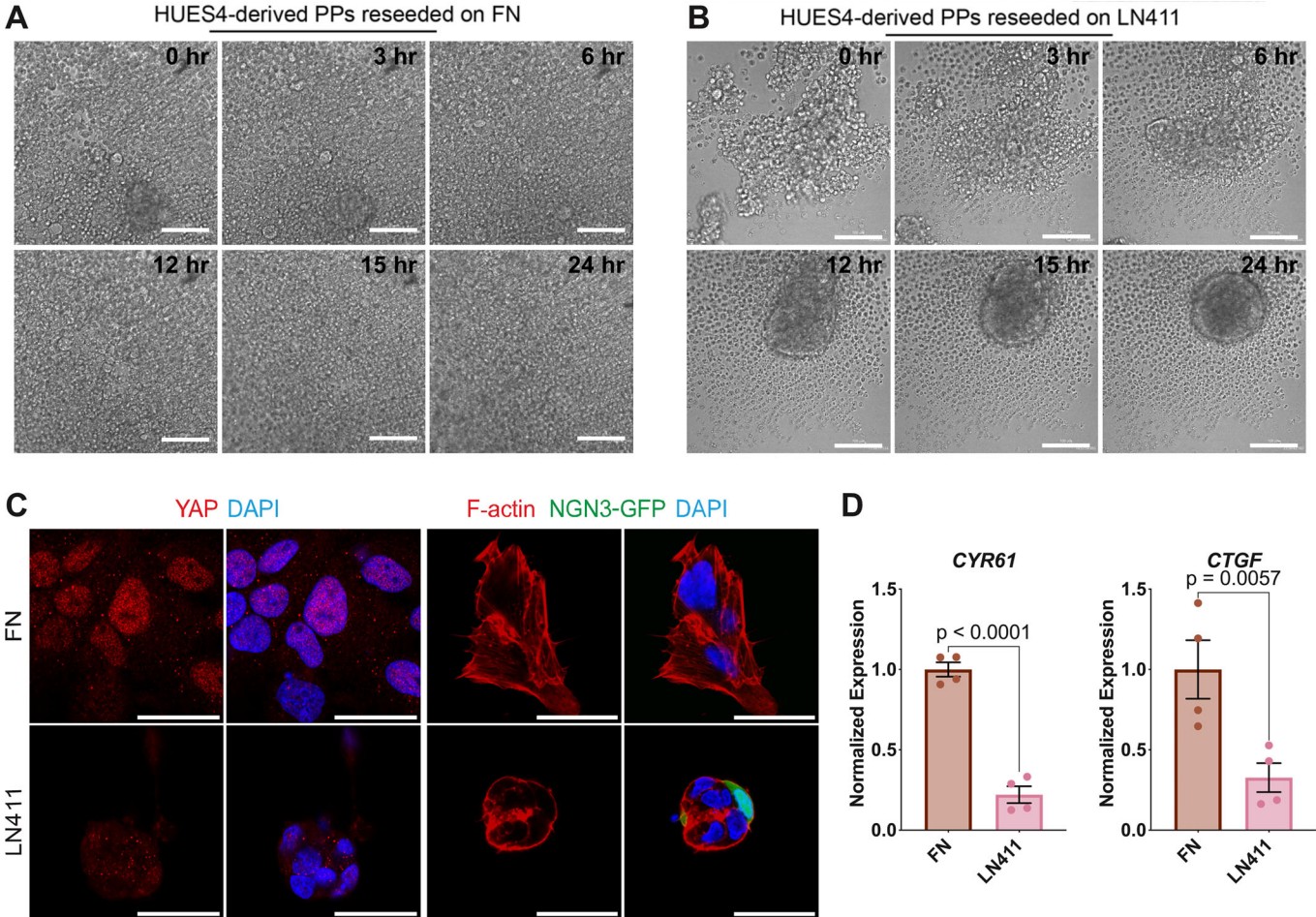

**Figure EV2. LN411 induces a confined morphology, reduces actin polymerization, and decreases YAP activity.**

(A) Live imaging of HUES4-derived pancreatic progenitors reseeded on FN by differential-interference contrast (DIC). Imaging started 1 h after reseeding. Scale bar = 100 μM. (B) Live imaging of HUES4-derived pancreatic progenitors reseeded on LN411 by differential-interference contrast (DIC). Imaging started 1 h after reseeding. Scale bar = 100 μM. (C) NGN3-GFP-derived pancreatic progenitors were sorted for GP2 using fluorescence-activated cell sorting, reseeded on FN (top panels) or LN411 (bottom panels) for 2 days and immunostained for YAP (left panels, red) or with phalloidin (right panels, red) in addition to DAPI (blue). Scale bar = 30 μm. (D) qPCR analysis of pancreatic of the YAP target genes CYR61 and CTGF in NGN3-GFP derived pancreatic progenitors 24 h after reseeding on LN411 or FN. The data were analyzed using two-tailed paired t-tests and are shown as mean expression ± SEM (n = 4, biological replicates).

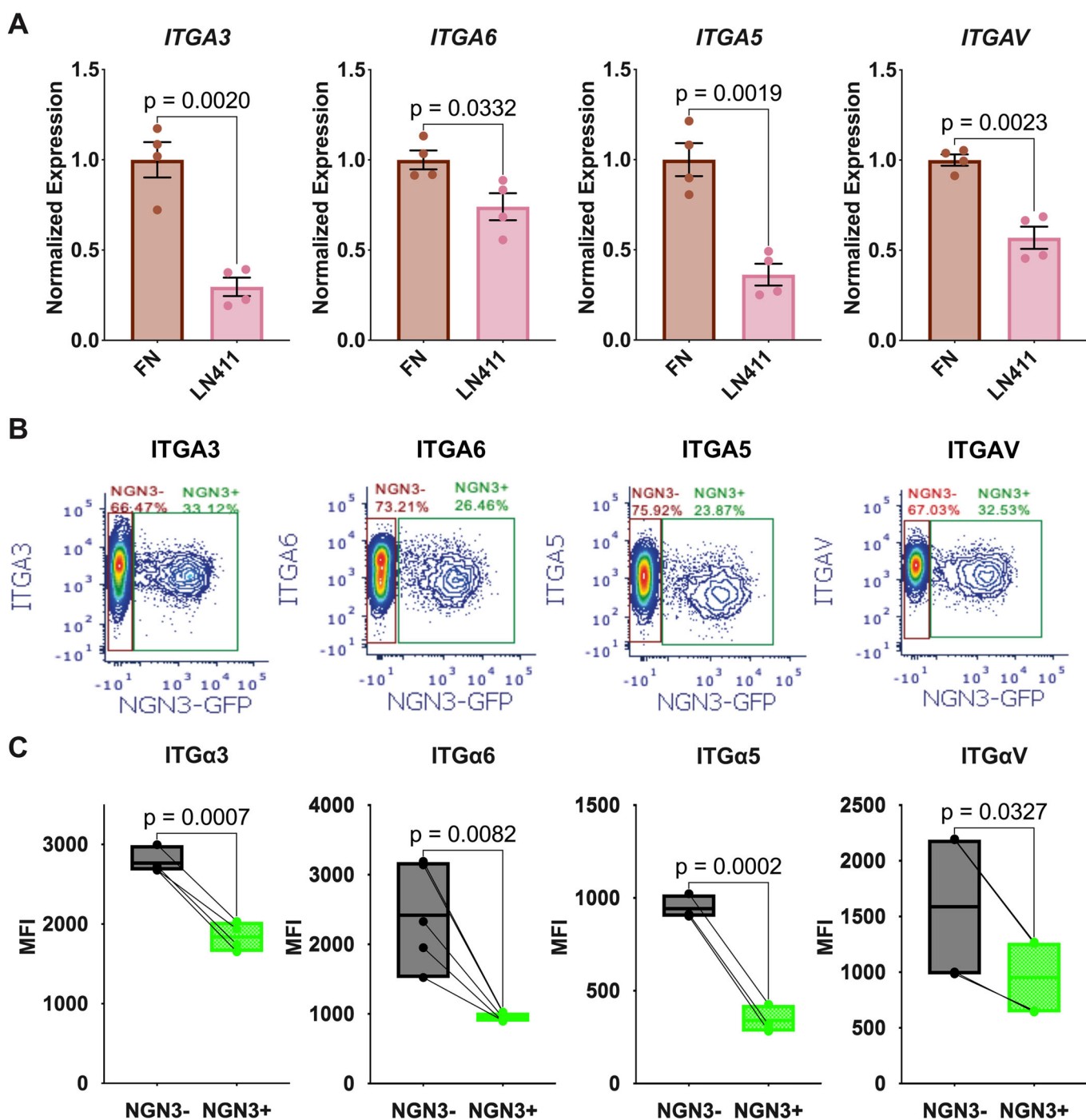

**Figure EV3. Reseeding on LN411 leads to downregulation of integrins.**

(A) qPCR analysis of indicated integrins in HUES4-derived pancreatic progenitors 4 days after reseeding on FN or LN411. The data were analyzed using two-tailed paired t-tests and are shown as mean expression ± SEM (n = 4, biological replicates). (B) Representative flow cytometry plots of Latrunculin B-treated NGN3-GFP-derived pancreatic progenitors immunostained for indicated integrins. (C) Quantification of mean fluorescence intensity in NGN3-GFP⁺ and NGN3-GFP⁻ cells as indicated in (B). Data were analyzed by two-tailed paired t- and are shown as boxplots with box boundaries extending to the minimum and maximum with a line at the mean (n = 4 for ITGα3 and ITGα6, n = 3 for ITGα5, n = 2 for ITGαV, biological replicates).

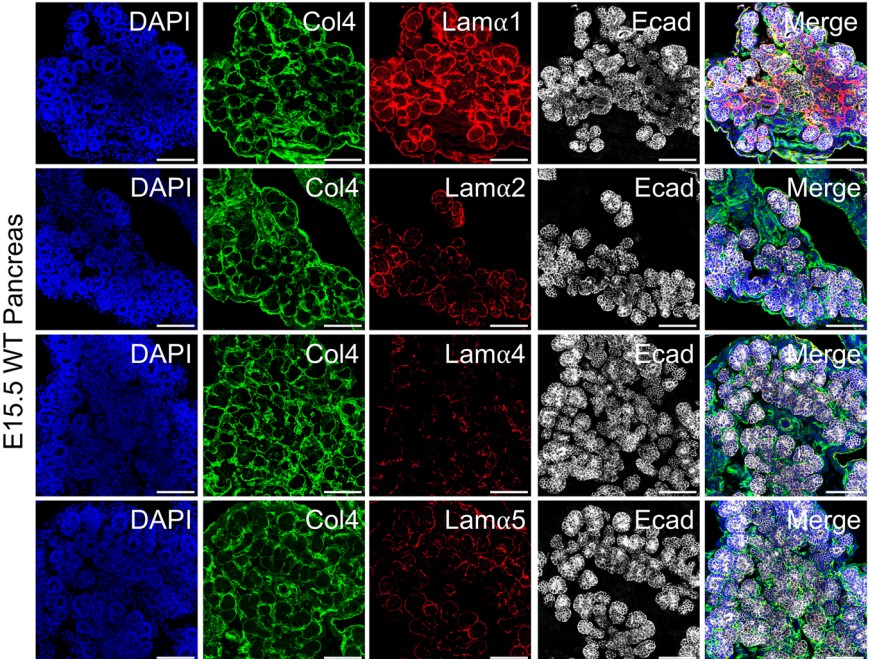

**Figure EV4. Heterogenous expression of laminin isoforms in the pancreatic epithelium.**

Immunofluorescent staining of 20-μm sections of E15.5 pancreas immunostained for collagen IV (Col4, green), E-cadherin (E-cad, white), and indicated laminin α chains (Lamα1-5, red), as well as DAPI (blue). Scale bar = 100 μM.

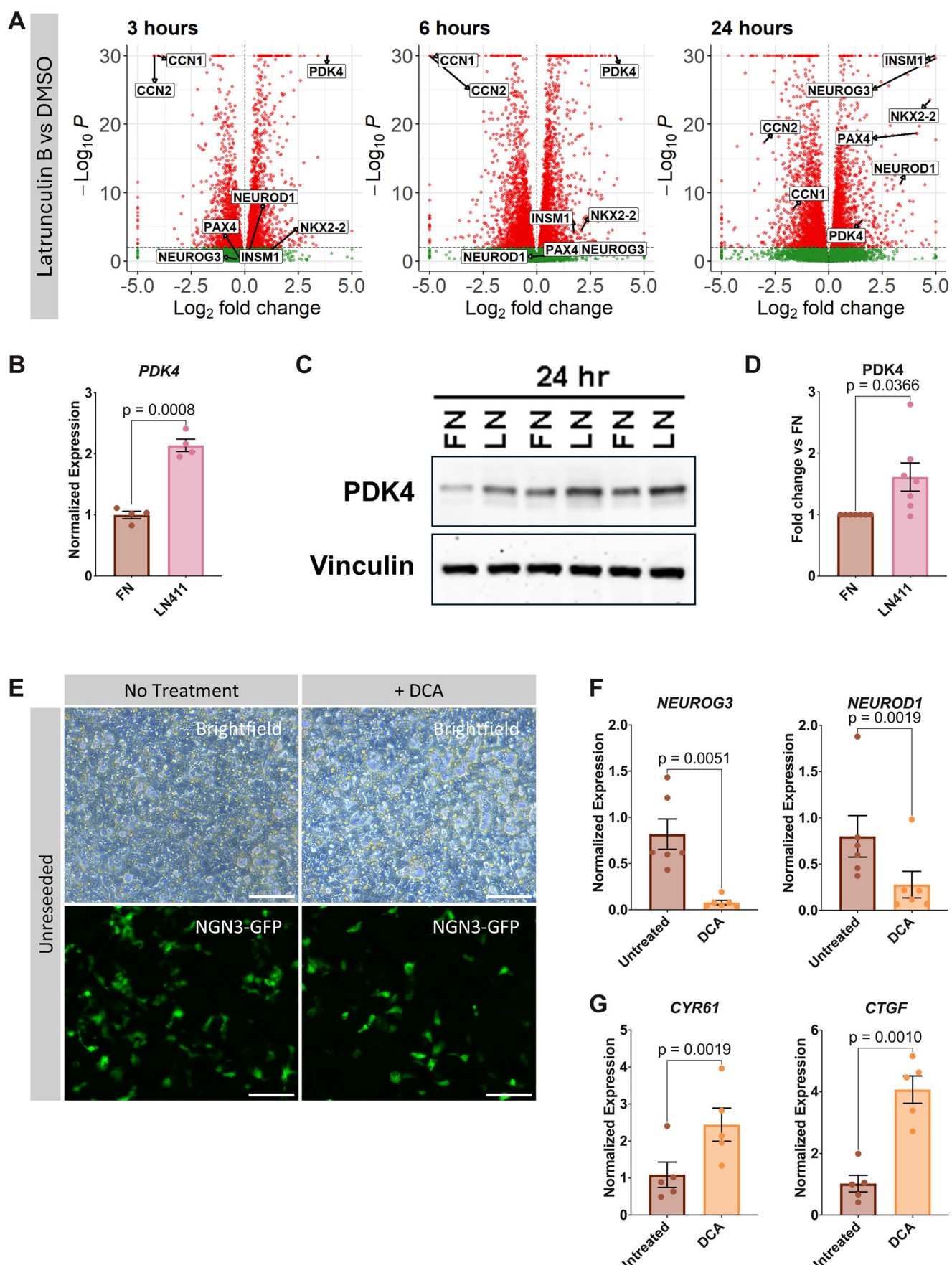

◀ **Figure EV5.** *PDK4 upregulation precedes endocrine specification in non-reseeded pancreatic progenitors.*

(A) Volcano plots of bulk RNA-seq analysis of NGN3-GFP-derived pancreatic progenitors treated with latrunculin B or DMSO at indicated time points. Expression of YAP target genes (*CCN1/CYR61, CCN2/CTGF*), early endocrine genes (*NEUROG3, NEUROD1, INSM1, NKX2-2*) and *PDK4* are indicated. Differential expression analysis was performed using DESeq2 ($n = 3$). (B) qPCR analysis of *PDK4* expression 24 h after reseeding on FN or LN411. The data were analyzed using two-tailed paired *t*-tests and are shown as mean expression ± SEM ($n = 4$). (C) Western blot of PDK4 expression 24 h after reseeding on FN or LN411 (LN). Vinculin is included as a loading control. (D) Quantification of (C). The data were analyzed using two-tailed paired *t*-tests and are shown as mean expression ± SEM ($n = 7$). (E) Brightfield (top panels) and epifluorescence (bottom panels) images of pancreatic progenitors after 2 days of no treatment or treatment with 10 mM DCA. Scale bar = 100 μm. (F) qPCR analysis of endocrine marker genes *NEUROG3* and *NEUROD1* in pancreatic progenitors after 2 days of no treatment or treatment with 10 mM DCA. The data were analyzed using two-tailed paired *t*-tests and are shown as mean expression ± SEM ($n = 6$). (G) qPCR analysis of YAP target genes *CYR61* and *CTGF* in pancreatic progenitors after 2 days of no treatment or treatment with 10 mM DCA. The data were analyzed using two-tailed paired *t*-tests and are shown as mean expression ± SEM ($n = 6$).

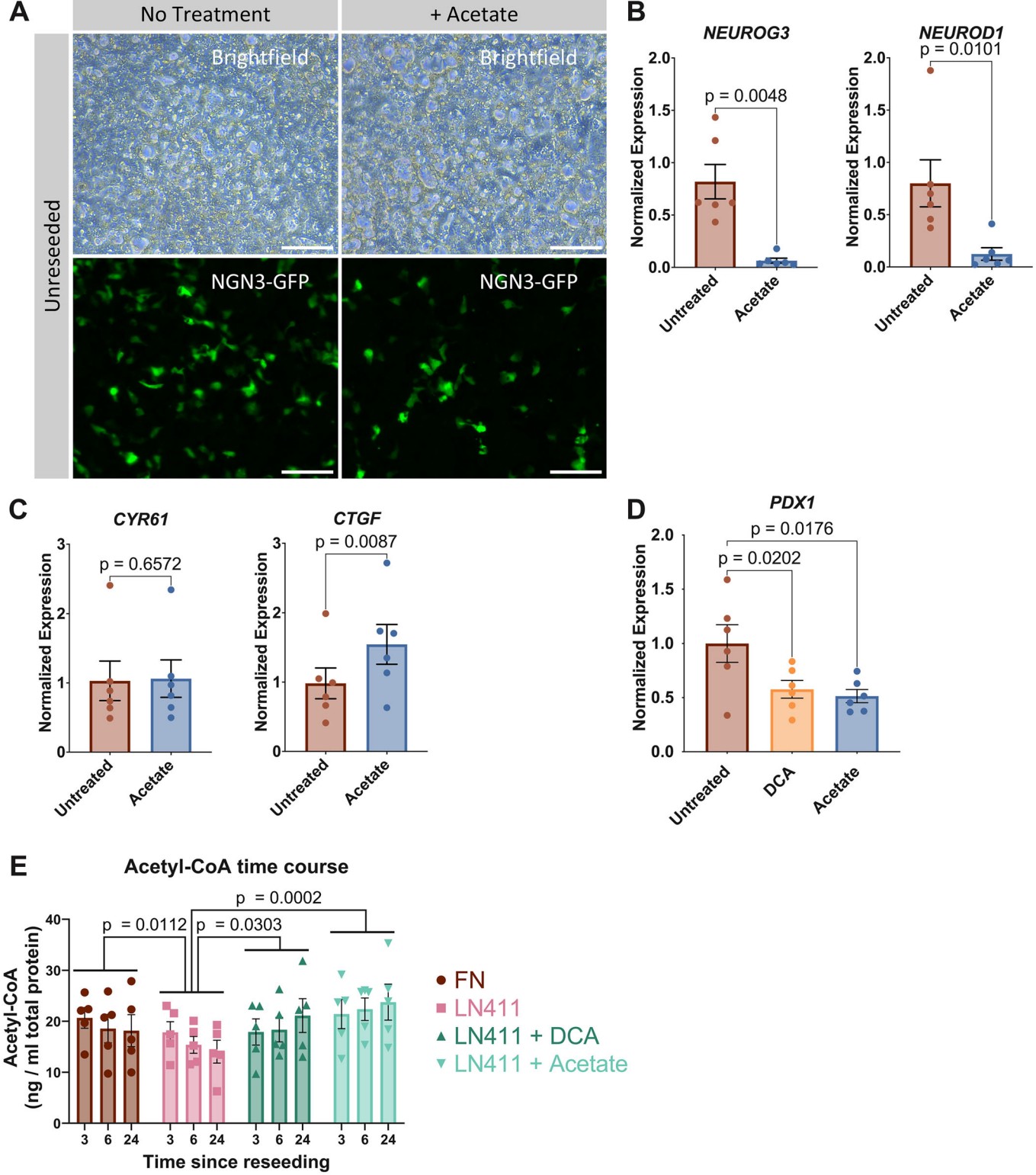

◀  **Figure EV6.  Acetate inhibits endocrine specification via a YAP-independent mechanism.**

(A) Brightfield (top panels) and epifluorescence (bottom panels) images of NGN3-GFP-derived pancreatic progenitors after 2 days of no treatment or treatment with 20 mM sodium acetate. Scale bar = 100 μm. (B) qPCR analysis of endocrine marker genes *NEUROG3* and *NEUROD1* in pancreatic progenitors after 2 days of no treatment or treatment with 20 mM sodium acetate. The data were analyzed using two-tailed paired *t*-tests and are shown as mean expression ± SEM (*n* = 6). (C) qPCR analysis of YAP target genes *CYR61* and *CTGF* in pancreatic progenitors after 2 days of no treatment or treatment with 20 mM sodium acetate. The data were analyzed using two-tailed paired *t*-tests and are shown as mean expression ± SEM (*n* = 6). (D) qPCR analysis of *PDX1* in pancreatic progenitors after 2 days of no treatment or treatment with 10 mM DCA or 20 mM sodium acetate. The data were analyzed by Dunnett's test with comparison to the untreated condition and are shown as mean expression ± SEM (*n* = 6). (E) ELISA analysis of intracellular acetyl-CoA concentrations in pancreatic progenitor cells at 3, 6, and 24 h after reseeding on FN, LN411, or LN411 with 10 mM DCA or 20 mM sodium acetate. The data were analyzed by two-way repeated measures ANOVA testing for treatment differences across the entire time course (3–24 h), followed by Dunnett's test with comparison to LN411 alone and are shown as mean expression ± SEM (*n* = 5).

**Figure EV7. PDK4 induces endocrine fate choice via both canonical and non-canonical functions.**

Schematic illustrating the proposed method of PDK4-mediated endocrine fate choice. Canonically, PDK4 blocks conversion of pyruvate to acetyl-CoA via inhibition of PDH (not shown). PDK4 additionally blocks YAP activity through a non-canonical mechanism. PDX1 is required for *NGN3* expression, while YAP inhibits *NGN3* expression. DCA targets PDK4 directly and thus increases YAP activity and decreases *PDX1* expression. Acetate is converted to Acetyl-CoA, bypassing the effect of PDK4. Acetate treatment thus reduces *PDX1* expression without affecting YAP activity.

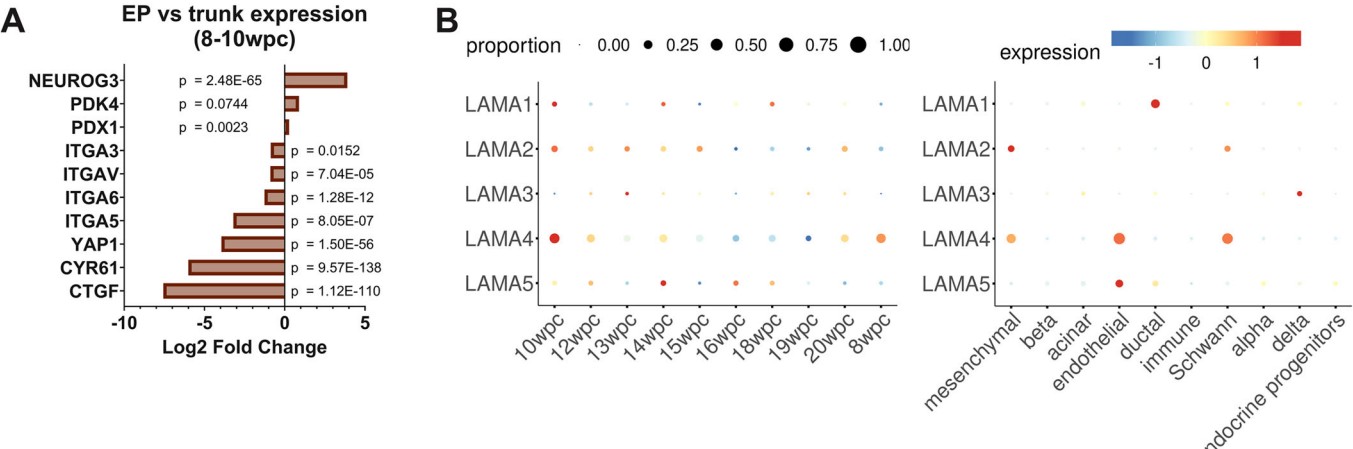

**Figure EV8. Single-cell RNA sequencing datasets validate the LN411-PDK4-endocrine specification pathway.**

(A) Log2 fold change in gene expression between endocrine progenitors (EP) and bipotent trunk progenitors (trunk) at 8–10 wpc in the OMIX001616 dataset (Ma et al, 2023). p-values are derived from MAST analysis (Finak et al, 2015) followed by Benjamini–Hochberg correction for multiple comparisons. (B) Dot plots of normalized expression of laminin α subunit genes in single-cell RNA sequencing samples of fetal pancreata according to developmental time (left) and cell type (right). Plots generated using humanpancreasdevelopment.org (Olaniru et al, 2023).

