## [Peer Review File · EMBO Reports]

Extracellular Matrix-Driven Metabolic Control of Pancreatic Endocrine Lineage Allocation

Christine Ebeid, Adam Rump, Chenglei Tian, Anant Mamidi, Adèle De Arcangelis, Gerard Gradwohl, and Henrik Semb

Corresponding author(s): Henrik Semb (henrik.semb@helmholtz-munich.de)

Review Timeline:

Transfer Date:	25th Mar 25
Editorial Decision:	4th Apr 25
Revision Received:	8th Jul 25
Editorial Decision:	10th Sep 25
Revision Received:	25th Sep 25
Accepted:	13th Oct 25

Transaction Report: This manuscript was transferred to EMBO reports following peer review at The EMBO Journal.

Referee #1:

In this study, Ebel et al. investigated the role of ECM and mechanical tension in the lineage specification of pancreatic endocrine cells. Using a combination of genetic approaches in vitro in human pluripotent stem cells and in vivo in the mouse, they identified Laminin LN411 as an extrinsic 'pro-endocrine' factor, which creates a local "soft" environment by reducing the expression of integrins $\alpha 3$ and $\alpha 6$ in bipotent endocrine-duct progenitor cells. Consistently, the in vitro knockdown in stem cells and in vivo knockout of these integrins lead to an increase of endocrine fate. The authors performed a time-course RNASeq to dissect the mechanisms downstream of LN411-induced endocrine specification and found the metabolic regulatory kinase PDK4. Based on a set of experiments in human pluripotent stem cells, PDK4 is proposed to act through both a YAP-dependent and YAP-independent metabolic mechanism, at the crossroad between metabolism and ECM-mediated mechanosignalling.

This is an interesting study which starts shedding light on an underexplored question: how mechanical and metabolic states influence cell fate decision. Specific concerns with the manuscript that need to be addressed to strengthen the conclusions drawn by the authors are listed below.

Main Concerns:

1) The systematic analysis of the different laminin isoforms showed that LN211 and LN411 possess similar endocrinogenic effects. It is unclear the rationale behind focusing only on LN411 and dropping LN211.

2) The images shown in Fig. EV4 do not support a heterogeneous expression of different laminin isoforms in the pancreatic epithelium, as stated by the authors; also, the conclusions are not fully consistent with previous literature. Laminins $\alpha 2$ and $\alpha 4$ have been actually reported as being the major laminins in the pancreatic acinar basement membrane (Miner et al 2004) and present in the blood vessels inside and around islets (Otonkoski et al 2008). The descriptive spatio-temporal analysis of the different laminin isoforms in vivo in pancreatic tissue in the mouse should be expanded and performed in human tissue, if available. Alternatively, publicly available scRNASeq datasets could be mined to assess which cell types in the pancreatic tissue express these different laminin isoforms. This would help better understanding their role and mechanism of action.

3) Were the NGN3-GFP cells shown in Fig. EV3B-C treated with Latrunculin B ? This should

be clearly stated in the Results and written in the figure legend. Also, the conclusion drawn from these experiments needs revision. The figure shows that the levels of expression of the integrins were reduced in NGN3-GFP+ endocrine cells compared to NGN3-GFP- fraction instead of the 'bipotent counterparts', as wrongly stated in the manuscript. There is no characterization of the NGN3-GFP- fraction, only the assumption that it corresponds to 'bipotent progenitors'.

4) The knockdown experiments of ITGA6 raise some concerns, mostly due to the minor effect of the targeting siRNA (see Fig. 2), which did not result into a statistically significant downregulation of ITGA6. Perhaps, CRISPR interference (CRISPRi) would be a more robust approach. Also, do the two integrins ITGA6 and ITGA3 compensate for each other in a human context? Finally, what are the consequences on the expression of ITGB1, the most abundant subunit in this epithelium, by the different laminin isoforms and upon ITGA3/6 knockdown?

5) The analysis of the mouse *Itga3/a6* knockout pancreas should be completed and the analysis of the phenotype carried out in vivo as much as possible. The authors claimed that this was not possible because of embryonic lethality but it is unclear at which stage and why/how this would hamper the dissection of the embryonic pancreata and their analysis. Anyway, the very preliminary analysis carried out on explants suggested an expansion of the endocrine compartment when compared to the pancreatic epithelium. The authors conclude that this might be due to accelerated endocrine specification and defective duct lineage, but this has not been tested here.

6) The role of PDK4 in endocrinogenesis is novel and interesting, however some of the findings in support of this conclusion are still preliminary. First, the analysis of the knockdown experiments using siRNA targeting PDK4 in hESCs should be expanded. For example, the requirement of PDK4 for PDX1 maintenance or for the regulation NKX6.1 and YAP should be tested in the PDK4-depleted cells instead of upon treatment with DCA or Acetate. These compounds exert multiple effects on pathways of intermediary metabolism and, therefore, have stronger and broader effects than just only on PDK4, possibly acting via other mechanisms.

Referee #2:

This is an interesting study that investigates the early cues required for specification of endocrine progenitors in the developing pancreas (the precursors of insulin-producing b-

islet cells). It is known that expression of the transcription factor Neurog3 plays a role in this process, which activates a Notch signalling pathway. Data for the authors' group further suggest involvement of YAP (upstream of Notch) and, hence, a mechano-transduction component. The authors here investigate the possibility that the ECM environment orchestrates these pathways, potentially providing the mechano-transduction component, through a novel PDK4 - pathway.

The focus is on laminins, the reason for which is not explained in the introduction but is presumably because laminins have long been implicated in insulin secretion by β -islet cells; human iPSC derived pancreatic endocrine progenitors are employed in in vitro studies.

The authors propose that specifically laminin 411 provides an early 'soft matrix' signal required for differentiation to endocrine progenitors, by down regulating integrin receptor that would normally promote adhesion to RGD containing matrices and some laminins and promoting expression of PDK4 and altering the metabolic status of the cells. The concept is interesting but the data provided do not support this hypothesis and several important controls are missing.

Critically, from the data provided it is not clear whether a laminin 411 containing basement membrane is ever in contact with the endocrine progenitors in vivo, nor whether simply laminin 411 reflects a soft matrix. Changes in mechanical signals provided by the ECM may indeed induced endocrine progenitor formation, however, whether laminin 411 provides this signal is not substantiated by the data shown, nor is the link between integrin expression changes and laminin 411 clear. The manuscript is therefore too preliminary for publication in EMBO J.

Major points

Overall in vitro experiments have a small number of replicates.

One point that is not clear is whether the 9 laminin isoforms tested in Fig.1 actually occur at the site of endocrine progenitor localization in vivo. From the data shown, it seems that only laminin a1 occurs in the ductal basement membrane, at least at the one time point shown. Conclusions cannot be made from the data provided for laminin a2, a4 or a5, as resolution is poor and magnifications are too low. Also, different developmental stages, corresponding to commitment of endocrine progenitors, are not shown. This is important information as from the model proposed, at some early time point in differentiation the

non-committed progenitor cells must come in contact with laminin 411, lose certain integrins and upregulate NGN3 and PDK4. Similarly, does in vivo integrin expression on progenitor cells show down regulation of integrins $\alpha 5$, αv , $\alpha 3$ and $\alpha 6$ coincident with their interaction with laminin 411?

The authors claim that laminin 411 results in reduced endocrine precursor cell adhesion- but this is not shown.

It is not clear to this reviewer, what the significance of the downregulation of αv , $\alpha 5$, $\alpha 3$ and $\alpha 6$ is. Do the authors propose that integrin $\alpha 6$ and $\alpha 3$ mediate binding to laminin 411 or to other laminins which may signal other signals preventing endocrine progenitor differentiation? This could be tested in adhesion/inhibition assays. The authors also fail to address another major laminin $\alpha 4$ receptor, MCAM, which mediates low affinity binding to this isoform only. What is its expression pattern during formation of endocrine progenitors, relative to laminin $\alpha 4$ expression? Are other major non-integrin receptors expression altered in the course of endocrine progenitor commitment, eg Lutheran (BCAM) and dystroglycan (the latter of which has also been implicated in events upstream of Notch signalling) ?

A caveat is that the laminin $\alpha 4$ knockout mouse, which was generated over 20 years ago, has no described pancreatic defect, although early stages of pancreas development have not been studied/reported. The absence of an overt pancreas effect in the Lama4^{-/-} mouse contrasts with the data reported in the manuscript (especially that of the integrin knockouts, if indeed they should represent laminin receptors). Additionally, although the authors claim that integrin $\alpha 3/\alpha 6$ double knockouts do not have an overt morphological defect in pancreas anlage, this is not reflected in the data shown, where ductal formation seems severely altered and the organ smaller. Further, it is important to show whether duct lineage markers are in fact being downregulated over the upregulation of endocrine lineage markers.

Specific comments regarding figs:

Fig 1 shows expression of glucagon and C-peptide by pancreatic progenitor cells plated on different laminins isoforms after 9 days of culture. However, the expression of these proteins before contact with laminins is not shown. This is important as the authors state in methods that the cells were maintained on laminin 521 prior to the experiment. Were the initial levels of glucagon and C-peptide similar among groups before plated on different laminins?

The authors further confirmed the endocrine commitment of cells by qPCR. Although

levels of NEUROG3 (used here as the major indicator of endocrine specification) was higher on cells plated on LM411 than on FN, it was only about 0.7 higher. To substantiate the data, qPCR showing downregulation of genes related to maintaining progenitor cells in an undifferentiated state, i.e., Hes1 (Notch signaling), would be important.

Addition points

The authors should consider that integrins alpha 5 and alpha v can also bind to laminin 511 - this has been shown by the researcher who has provided the laminin antibodies to the authors. There is also no convincing data that integrin alpha3beta1 can bind to laminin 411. This fact should be taken into consideration (or indeed tested in in vitro adhesion assays using the cells employed in the study).

Fig 3/ Supplementary fig. EV4 The authors showed in vivo expression of laminin alpha4 and endocrinogenesis. The immunofluorescence images shown in Fig. EV4 are inconclusive and do not support the statement "which revealed a heterogeneous distribution of laminin alpha4" (line 132-133). In fact, laminin alpha4 was previously described to be continuously expressed on the vasculature basement membranes (Korpos et al, Diabetes 62:531-542, 2013).

Please specify the scale bar sizes in the legend or increase the size of those in the images.

The authors should revisit the discussion section, where some references are not correct; for instance, authors refer to "Mahmoud, 2023" (a review article) when discussing cell fate decisions and cell-ECM interactions (lines 255-256), where the experimental paper that first described such interactions is important to cite.

Furthermore, the downregulation of integrins by laminin 411 coating is vaguely discussed. Such an event, presented as an important part of the manuscript, should be better discussed. Does this down regulation of integrins lead to loss of adhesion of the cells to the substrate; is loss of adhesion per se a signal for differentiation of endocrine progenitors. In the discussion, it would be beneficial to address whether the downregulation of integrins by cells upon contact with laminins was reported previously, in particular in other endocrine systems.

The authors could consider further discussion on pancreas blood vessels (BV) and its involvement in the the development of the endocrine pancreas. The function of BVs in pancreas is mentioned in the discussion and that the literature so far is rather unclear, however, for this report in which BV basement membranes probably contribute

substantially to the events presented, it's potential influence could be discussed.

Authors mix gene names with protein names- Lama4, Lama1, etc are gene names, written in Italics, the proteins are laminin a4, laminin a1 etc. This should be corrected in figures and the text, as it leads to confusion.

Statistics are missing in some graphs from Fig 2B.

Referee #3:

In this manuscript, Ebeid et al. investigate how extracellular matrix (ECM) cues, specifically low-adhesion laminin 411 (LN411), influence pancreatic endocrine lineage allocation. The authors propose that LN411 reduces integrin $\alpha 3$ and $\alpha 6$ expression, thereby lowering cell-ECM adhesion and mechanical tension, which leads to upregulation of PDK4. PDK4 is suggested to promote endocrine specification through two distinct mechanisms: (1) non-canonical inhibition of YAP signaling and (2) canonical reduction of acetyl-CoA to sustain PDX1 expression.

The study addresses a significant biological question at the intersection of ECM signaling, mechanotransduction, metabolism, and endocrine fate choice, with potential relevance for understanding pancreas development and improving in vitro differentiation protocols for diabetes therapy.

While the study presents interesting and novel connections, particularly the involvement of PDK4 as a mechanosensitive metabolic regulator, several issues limit the strength and novelty of the conclusions. The mechanistic pathways are not fully developed, and in vivo validation is limited, raising concerns about the physiological relevance of the proposed mechanism. Furthermore, some aspects of the manuscript are not well articulated, and the focus on PDK4 may overlook other relevant pathways suggested by the authors' RNA-seq data.

Overall, while the study has potential value, I believe major revisions are needed to fully support the conclusions and to improve the clarity and impact of the manuscript.

Major Concerns

Incomplete mechanistic understanding of PDK4's dual role (YAP inhibition and PDX1 maintenance):

The proposed non-canonical inhibition of YAP by PDK4 is central to the model but lacks mechanistic evidence. It remains unclear whether this occurs through known intermediates (e.g., Septin-2) or other unknown substrates. Similarly, the claim that PDK4-mediated reduction of acetyl-CoA maintains PDX1 expression is speculative and lacks direct metabolite measurements (e.g., acetyl-CoA levels, FOXA2 acetylation status). The authors should provide additional experimental data, or at least address these gaps clearly in the discussion and propose a mechanistic hypothesis based on existing literature.

Limited in vivo validation and physiological relevance:

The study relies primarily on in vitro systems and ex vivo explants, with no direct in vivo evidence of LN411's role in endocrine specification. The spatial relationship between LN411 deposition and NEUROG3+ progenitors during pancreas development is not clearly shown. Additional data (e.g., in vivo co-localization of LN411 and endocrine progenitors, conditional models) or a more thorough discussion of these limitations is necessary to validate the proposed mechanism in a physiological context.

Selective focus on PDK4 without addressing other candidates from RNA-seq data: The RNA-seq analysis identifies other significantly upregulated genes (e.g., FOXA3, RFX6, FOXP1, DDIT4), some of which are known to influence pancreatic development. The authors should discuss these additional factors, and whether endocrine induction may involve a multifactorial response rather than a PDK4-centric pathway. This would provide a more balanced interpretation of their findings.

Articulation and clarity of the proposed model: The manuscript contains dense and sometimes confusing descriptions of mechanisms, especially regarding the dual role of PDK4. The authors should revise key sections for clarity and logical flow, and clearly define terms such as "canonical" and "non-canonical" early in the manuscript. Including a schematic summary figure of the proposed mechanism would help guide the reader.

Minor Concerns

Functional application in hPSC-derived beta-cell differentiation: Given the potential translational relevance, exploring whether LN411/PDK4 modulation could improve beta-cell differentiation protocols would enhance the impact of the study. Even preliminary data could provide an interesting proof-of-concept.

Further exploration of PDK4's non-canonical effects: Investigating potential interacting proteins or signaling partners of PDK4 that may mediate YAP inhibition (e.g., via proteomics or proximity-labeling approaches) would add mechanistic depth.

Consideration of temporal dynamics: Time-course experiments measuring acetyl-CoA, YAP localization, PDK4 expression, and NEUROG3 induction could clarify the sequence of events and strengthen the causality of the proposed pathway.

Based on the above points, I recommend Major Revision if this manuscript is being considered for EMBO, as it addresses a novel and interesting aspect of pancreas development that could be suitable with significant clarification and additional data or discussion. However, in my view, this manuscript may not yet meet the standard for EMBO Journal, as it requires substantially deeper mechanistic and in vivo validation to reach the high bar of conceptual and technical advance expected there.

Dear Prof. Semb

Thank you for the transfer of your research manuscript and the associated referee reports to our journal.

As my colleague from The EMBO Journal, Ieva Gailite, informed you, we are interested in publishing your manuscript at EMBO Reports. I repeat below the points we considered important to address for potential publication at our journal (point 1-4).

Thank you for providing a preliminary point-by-point response. I suggest that you revise your study along the lines proposed in your rebuttal.

Points 1-4:

- 1) The focus on LN411 over LN211 as well as the focus on the PDK4 pathway instead of other DFE genes should be clearly articulated, but it is not essential to expand the analysis to other potential candidates from the RNA-seq analysis.
- 2) Further mechanistic understanding is not required, but the suggestion from referee 3 to analyse the temporal dynamics of acetyl-CoA, YAP localization, PDK4 expression, and NEUROG3 induction should be considered, as it would strengthen the causality of the proposed pathway. This time-course analysis, including a measurement of acetyl-CoA levels could also bolster the hypothesis that PDK4-mediated reduction of acetyl-CoA maintains PDX1 expression.
- 3) A major concern relates to the specificity of the proposed LN411 - integrin $\alpha 3/\alpha 6$ interaction and whether non-committed progenitor cells come in contact with laminin 411 in vivo. At the least, the spatio-temporal expression of different laminins and integrins as well as the expression of NGN3 and PDK4 in vivo should be expanded, as suggested by referee 2, major point 2 and referee 1, point 2. Available scRNA-seq datasets could be mined as well.
- 4) Referee 2 also raises the point whether laminin $\alpha 4$ signals via integrin $\alpha 3/\alpha 6$ or whether other known receptors, such as MACM. And vice versa, whether integrin $\alpha 3/\alpha 6$ LOF affects binding to other laminins in addition to laminin $\alpha 4$. These points could be addressed in the in vitro setting.

Please address all referee concerns in a complete point-by-point response. Acceptance of the manuscript will depend on a positive outcome of a second round of review. It is EMBO Reports policy to allow a single round of revision only and acceptance or rejection of the manuscript will therefore depend on the completeness of your responses included in the next, final version of the manuscript.

We realize that it is difficult to revise to a specific deadline. In the interest of protecting the conceptual advance provided by the work, we recommend a revision within 3 months (July 4th). Please discuss the revision progress ahead of this time with the editor if you require more time to complete the revisions.

I am also happy to discuss the revision further via e-mail or a video call, if you wish.

=====
IMPORTANT NOTE:

We perform an initial quality control of all revised manuscripts before re-review. Your manuscript will FAIL this control and the handling will be delayed IN CASE the following APPLIES:

- 1) A data availability section providing access to data deposited in public databases is missing. If you have not deposited any data, please add a sentence to the data availability section that explains that.
- 2) Your manuscript contains statistics and error bars based on $n=2$. Please use scatter blots in these cases. No statistics should be calculated if $n=2$.

=====

2) individual production quality figure files as .eps, .tif, .jpg (one file per figure).

Please download our Figure Preparation Guidelines (figure preparation pdf) from our Author Guidelines pages <https://www.embopress.org/page/journal/14693178/authorguide> for more info on how to prepare your figures.

4) a complete author checklist, which you can download from our author guidelines

(<<https://www.embopress.org/page/journal/14693178/authorguide>>). Please insert information in the checklist that is also reflected in the manuscript. The completed author checklist will also be part of the RPF.

5) Please note that all corresponding authors are required to supply an ORCID ID for their name upon submission of a revised manuscript (<<https://orcid.org/>>). Please find instructions on how to link your ORCID ID to your account in our manuscript tracking system in our Author guidelines

(<<https://www.embopress.org/page/journal/14693178/authorguide#authorshipguidelines>>)

6) We replaced Supplementary Information with Expanded View (EV) Figures and Tables that are collapsible/expandable online. A maximum of 5 EV Figures can be typeset. EV Figures should be cited as "Figure EV1, Figure EV2" etc... in the text and their respective legends should be included in the main text after the legends of regular figures.

7) Before submitting your revision, primary datasets (and computer code, where appropriate) produced in this study need to be deposited in an appropriate public database (see <

<https://www.embopress.org/page/journal/14693178/authorguide#dataavailability>>).

The accession numbers and database should be listed in a formal "Data Availability " section (placed after Materials & Method) that follows the model below (see also < <https://www.embopress.org/page/journal/14693178/authorguide#dataavailability>>). Please note that the Data Availability Section is restricted to new primary data that are part of this study.

Data availability

Additional information on source data and instruction on how to label the files are available

<<https://www.embopress.org/page/journal/14693178/authorguide#sourcedata>>.

10) Figure legends and data quantification:

- the name of the statistical test used to generate error bars and P values,
 - the EXACT p-values,
 - the number (n) of independent experiments (please specify technical or biological replicates) underlying each data point,
 - the nature of the bars and error bars (s.d., s.e.m.)
-
- If the data are obtained from n {less than or equal to} 5, show the individual data points in addition to the SD or SEM.
 - If the data are obtained from n {less than or equal to} 2, use scatter blots showing the individual data points.

11) Our journal encourages inclusion of *data citations in the reference list* to directly cite datasets that were re-used and obtained from public databases. Data citations in the article text are distinct from normal bibliographical citations and should directly link to the database records from which the data can be accessed. In the main text, data citations are formatted as follows: "Data ref: Smith et al, 2001" or "Data ref: NCBI Sequence Read Archive PRJNA342805, 2017". In the Reference list, data citations must be labeled with "[DATASET]". A data reference must provide the database name, accession number/identifiers and a resolvable link to the landing page from which the data can be accessed at the end of the reference. Further instructions are available at <<https://www.embopress.org/page/journal/14693178/authorguide#referencesformat>>.

12) All Materials and Methods need to be described in the main text using our 'Structured Methods' format. According to this format, the Methods section includes a Reagents and Tools Table (listing key reagents, experimental models, software and relevant equipment and including their sources and relevant identifiers) followed by a Methods and Protocols section describing the methods, ideally using a step-by-step protocol format. The aim is to facilitate adoption of the methodologies across labs. Please download and fill our Reagents and Tools Table template (.docx), which you can find in our author guidelines:

13) As part of the EMBO publication's Transparent Editorial Process, EMBO Reports publishes online a Review Process File to accompany accepted manuscripts. This File will be published in conjunction with your paper and will include the referee reports, your point-by-point response and all pertinent correspondence relating to the manuscript.

Yours sincerely,

We are grateful for the constructive feedback on our manuscript "Extracellular Matrix-Driven Metabolic Control of Pancreatic Endocrine Lineage Allocation." We have carefully addressed all points raised and believe the manuscript is significantly strengthened as a result. Below, we provide a detailed response to each of your editorial recommendations:

1) The focus on LN411 over LN211 as well as the focus on the PDK4 pathway instead of other DFE genes should be clearly articulated, but it is not essential to expand the analysis to other potential candidates from the RNA-seq analysis.

We have substantially revised the manuscript to provide clear rationale for our focus on LN411 and PDK4:

LN411 vs LN211: We have updated lines 92-95 to explicitly state that while both LN211 and LN411 showed endocrinogenic effects in the HUES4 cell line, the effect of LN211 was cell line-specific. Only LN411 demonstrated conserved endocrinogenic effects across multiple hESC lines, including the independent SA121-derived NGN3-GFP reporter line. This cell line-independent effect provided strong justification for focusing our mechanistic studies on LN411.

PDK4 selection: We have revised lines 174-183 to more clearly articulate why PDK4 was selected from the 9 overlapping genes between LN411 and latrunculin B treatments. We now explicitly state that the known functions of FST, RFX6, RGS4, and FOXP1 make them unlikely candidates for mediating the endocrinogenic effects. Specifically:

- FST has been shown to downregulate rather than upregulate endocrine induction (Miralles et al., 1998)
- RFX6, RGS4, and FOXP1 function downstream of NEUROG3 in endocrine development, delamination, and α -cell function, respectively (Serafimidis et al., 2011; Smith et al., 2010; Spaeth et al., 2015)
- In contrast, DDIT4, MAN1A1, and PDK4 have not been previously implicated in pancreas development, while PDK4's established role in ECM attachment (Grassian et al., 2011) and anchorage-independent growth (Tambe et al., 2019) makes it a compelling candidate for further investigation.

2) Further mechanistic understanding is not required, but the suggestion from referee 3 to analyse the temporal dynamics of acetyl-CoA, YAP localization, PDK4 expression, and NEUROG3 induction should be considered, as it would strengthen the causality of the proposed pathway. This time-course analysis, including a measurement of acetyl-CoA levels could also bolster the hypothesis that PDK4-mediated reduction of acetyl-CoA maintains PDX1 expression.

We have comprehensively addressed the temporal dynamics as suggested:

Enhanced temporal analysis: We have updated Figure 4A to include PDX1 expression data and revised the text (lines 188-190 and 225-227) to emphasize the temporal patterns revealed by our time-course RNA sequencing.

Figure 4. LN411 induces endocrine specification via PDK4

(A) Volcano plots of bulk RNA-seq analysis of NGN3-GFP-derived pancreatic progenitors reseeded on LN411 or FN at indicated time points. Expression of YAP target genes (*CCN1/CYR61*, *CCN2/CTGF*), early endocrine genes (*NEUROG3*, *NEUROD1*, *INSM1*, *NKX2-2*), *PDX1* and *PDK4* are indicated.

We now highlight the coordinated upregulation of PDK4 and PDX1 alongside downregulation of YAP targets CCN1 and CCN2 at the 6-hour time point, which precedes significant upregulation of endocrine specification genes, including NEUROG3.

Acetyl-CoA measurements: We have added new experimental data analyzing acetyl-CoA dynamics upon exposure to FN, LN411 alone or LN411 in combination with DCA or acetate (new Figure EV6E, lines 227-230).

Figure EV6. Acetate inhibits endocrine specification via a YAP-independent mechanism.

(E) ELISA analysis of intracellular acetyl-CoA concentrations in pancreatic progenitor cells at 3, 6, and 24 hours after reseeded on FN, LN411, or LN411

with 10mM DCA or 20mM sodium acetate. The data were analyzed by two-way repeated measures ANOVA testing for treatment differences across the entire time course (3-24h), followed by Dunnett's test with comparison to LN411 alone and are shown as mean expression \pm SEM (n = 5).

These data directly demonstrate:

- Reduced intracellular acetyl-CoA levels upon LN411 exposure (as compared to FN)
- The LN411-induced reduction is counteracted by concurrent DCA or acetate treatment
- The LN411-induced reduction of intracellular acetyl-CoA precedes the observed downregulation of PDX1 at 48 hours

These temporal relationships strongly support our model that PDK4-mediated reduction of acetyl-CoA maintains PDX1 expression during endocrine specification.

3) A major concern relates to the specificity of the proposed LN411 - integrin α 3/ α 6 interaction and whether non-committed progenitor cells come in contact with laminin 411 *in vivo*. At the least, the spatio-temporal expression of different laminins and integrins as well as the expression of NGN3 and PDK4 *in vivo* should be expanded, as suggested by referee 2, major point 2 and referee 1, point 2. Available scRNA-seq datasets could be mined as well.

Due to the highly dynamic nature of the developing pancreatic epithelium (Nyeng et al., 2019), a complete assessment of spatiotemporal ECM-cell interactions would require advanced live cell imaging with multi-colour mouse reporter lines. Static images could be misleading as NEUROG3+ cells undergo significant cell migration. However, we have extensively mined available scRNA-seq datasets to strengthen the *in vivo* relevance of our findings:

Human fetal pancreas analysis: We analyzed published scRNA-seq data focusing on 8-10 weeks post-conception (wpc), which corresponds to the onset of human endocrinogenesis. We specifically chose this timepoint because:

- It contains well-defined bipotent progenitor and endocrine progenitor populations
- Later timepoints lack clear intermediate endocrine-specified cells and defined bipotent progenitors

Our analysis (new Figure EV7A) reveals that the expression pattern perfectly aligns with our model:

Figure EV7. Single cell RNA sequencing datasets validate the LN411-PDK4-endocrine specification pathway.

(A) Log2 fold change in gene expression between endocrine progenitors (EP) and bi-potent trunk progenitors (trunk) at 8-10wpc in the OMIX001616 dataset (Ma *et al*, 2023). p-values are derived from MAST analysis (Finak *et al*, 2015) followed by Benjamini-Hochberg correction for multiple comparisons.

Specifically:

- ITGA3, ITGA6, ITGA5, and ITGAV are all significantly downregulated
- PDK4 shows upregulation (though not reaching statistical significance, likely due to detection in a small fraction of cells)
- YAP target genes show the expected downregulation
- PDX1 is significantly upregulated in endocrine progenitors

*Laminin expression analysis:* Using the comprehensive (Olaniru et al., 2023) dataset, we assessed LAMA subtype expression across human pancreatic development (new Figure EV7B):

Figure EV7. Single cell RNA sequencing datasets validate the LN411-PDK4-endocrine specification pathway.

(B) Dot plots of normalized expression of laminin α subunit genes in single-cell RNA sequencing samples of fetal pancreata according to developmental time (left) and cell type (right). Plots generated using humanpancreasdevelopment.org (Olaniru *et al*, 2023).

This analysis specifically shows that:

- LAMA4 expression peaks at 8-10 wpc, precisely coinciding with endocrine specification onset
- LAMA4 is predominantly expressed by non-epithelial cells, particularly Schwann cells and endothelial cells
- This pattern supports our model of heterogeneous LN411 deposition from surrounding mesenchyme driving local endocrine fate decisions

We have added discussion of these findings on lines 235-256 and 286-299, emphasizing how the *in vivo* expression patterns validate our proposed LN411-PDK4 pathway in human pancreatic development.

4) Referee 2 also raises the point whether laminin α 4 signals via integrin α 3/ α 6 or whether other known receptors, such as MACM. And vice versa, whether integrin α 3/ α 6 LOF affects binding to other laminins in addition to laminin α 4. These points could be addressed in the *in vitro* setting.

We appreciate this important question and recognize we may not have been sufficiently clear about our proposed mechanism in the original manuscript. We would like to clarify several key points:

Our mechanistic model: We do not claim that LN411 signals through integrins α 3/ α 6. Rather, our data demonstrate that LN411 exposure leads to a coordinated downregulation of multiple integrins (α 3, α 6, α 5, and α V) at both RNA and protein levels (Figures 2A-B, EV3A). This global reduction in integrin expression, not specific LN411-integrin interactions, drives the downstream effects on endocrine specification. This matches our previous work, which shows that knockdown of *ITGA5* is sufficient to induce increased endocrine specification *in vitro* (Mamidi *et al.*, 2018).

Mechanism of action: Our model proposes that LN411, as a low-adhesion ECM component (Ishikawa *et al.*, 2014), triggers integrin internalization through disrupted focal adhesion and stress fiber formation (Du *et al.*, 2011). This is fundamentally different from classical integrin-mediated signaling. The reduction in total integrin levels, rather than specific receptor-ligand interactions, creates the "soft" mechanical environment that promotes endocrine specification.

Integrin knockdown experiments: Our siRNA experiments (Figures 2C-D) and in vivo knockout studies (Figure 3) demonstrate that reducing integrin $\alpha 3/\alpha 6$ levels is sufficient to enhance endocrine specification, regardless of which ECM components are present. This supports our model that the key mechanism is reduced cell-ECM adhesion overall, not specific laminin-integrin pairings.

Addressing the specific questions:

- Regarding other LN411 receptors (e.g., MCAM): While we cannot exclude that LN411 may interact with other receptors, our data clearly show that the endocrinogenic effect is mediated through reduced integrin expression and decreased cell-ECM adhesion, as evidenced by the phenocopy achieved through integrin knockdown alone (Fig. 2C-D). We have previously demonstrated this effect for *ITGA5*, an integrin that does not bind LN411 (Mamidi et al., 2018).
- Regarding integrin $\alpha 3/\alpha 6$ binding to other laminins: This is precisely our point - reducing integrin $\alpha 3/\alpha 6$ levels decreases overall cell-ECM adhesion regardless of the specific laminin isoforms present, which is why integrin knockdown on LN521 (which does not induce integrin downregulation) can mimic the effect of LN411 exposure.

We have clarified these points in the revised manuscript (lines 124-125 and 275-277) to prevent future misunderstanding of our proposed mechanism.

Other changes:

In light of our new findings, we have updated lines 231-234 and 257-262 to more clearly state an overview of our findings. We have fixed typos on lines 164, 319, 324. We have updated figure texts on lines 408, 587-592, and 596-606.

To comply with the Structured Methods format, we have moved the table of reagents to a separate document. We have updated methods lines 636-637, 677-678, 689, 692, 702, 710, 712, and 728 to fix typos and formatting issues. We have added lines 747-763 to describe the added methods. We have further revised our Statistical analysis section on lines 767-772 to clarify our exact statistical approach.

Reviewer 1

In this study, Ebeld et al. investigated the role of ECM and mechanical tension in the lineage specification of pancreatic endocrine cells. Using a combination of genetic approaches in vitro in human pluripotent stem cells and in vivo in the mouse, they identified Laminin LN411 as an extrinsic 'pro-endocrine' factor, which creates a local "soft" environment by reducing the expression of integrins $\alpha 3$ and $\alpha 6$ in bipotent endocrine-duct progenitor cells. Consistently, the in vitro knockdown in stem cells and in vivo knockout of these integrins lead to an increase of endocrine fate. The authors performed a time-course RNASeq to dissect the mechanisms downstream of LN411-induced endocrine specification and found the metabolic regulatory kinase PDK4. Based on a set of experiments in human pluripotent stem cells, PDK4 is proposed to act through both a YAP-dependent and YAP-independent metabolic mechanism, at the crossroad between metabolism and ECM-mediated mechanosignalling.

This is an interesting study which starts shedding light on an underexplored question: how mechanical and metabolic states influence cell fate decision. Specific concerns with the manuscript that need to be addressed to strengthen the conclusions drawn by the authors are listed below.

Main Concerns:

1) The systematic analysis of the different laminin isoforms showed that LN211 and LN411 possess similar endocrinogenic effects. It is unclear the rationale behind focusing only on LN411 and dropping LN211.

Response: We have revised the manuscript (lines 92-95) to explicitly state that while both LN211 and LN411 showed endocrinogenic effects in the HUES4 cell line, the effect of LN211 was cell line specific. Only LN411 demonstrated conserved endocrinogenic effects across multiple hESC lines, including the independent SA121-derived NGN3-GFP reporter line. This cell line-independent effect provided strong justification for focusing our mechanistic studies on LN411.

2) The images shown in Fig. EV4 do not support a heterogenous expression of different laminin isoforms in the pancreatic epithelium, as stated by the authors; also, the conclusions are not fully consistent with previous literature. Laminins $\alpha 2$ and $\alpha 4$ have been actually reported as being the major laminins in the pancreatic acinar basement membrane (Miner et al 2004) and present in the blood vessels inside and around islets (Otonkoski et al 2008). The descriptive spatio-temporal analysis of the different laminin isoforms in vivo in pancreatic tissue in the mouse should be expanded and performed in

human tissue, if available. Alternatively, publicly available scRNASeq datasets could be mined to assess which cell types in the pancreatic tissue express these different laminin isoforms. This would help better understanding their role and mechanism of action.

Response: Using the comprehensive dataset (Olaniru et al, 2023), we assessed LAMA subtype expression across human pancreatic development (new Fig. EV8B).

This analysis specifically shows that:

- LAMA4 expression peaks at 8-10 wpc, precisely coinciding with endocrine specification onset.
- LAMA4 is predominantly expressed by non-epithelial cells, particularly Schwann cells and endothelial cells.
- This pattern supports our model of heterogeneous LN411 deposition from surrounding mesenchyme driving local endocrine fate decisions.

Figure EV8. Single cell RNA sequencing datasets validate the LN411-PDK4-endocrine specification pathway.

(B) Dot plots of normalized expression of laminin α subunit genes in single-cell RNA sequencing samples of fetal pancreata according to developmental time (left) and cell type (right). Plots generated using humanpancreasdevelopment.org (Olaniru et al, 2023).

3) Were the NGN3-GFP cells shown in Fig. EV3B-C treated with Latrunculin B? This should be clearly stated in the Results and written in the figure legend. Also, the conclusion drawn from these experiments needs revision. The figure shows that the levels of expression of the integrins were reduced in NGN3-GFP+ endocrine cells compared to NGN3-GFP- fraction instead of the 'bipotent counterparts', as wrongly stated in the manuscript. There is no characterization of the NGN3-GFP- fraction, only the assumption that it corresponds to 'bipotent progenitors'!

Response: The cells shown in Fig. EV3B-C were treated with Latrunculin B, as stated on line 111. We have further clarified the use of Latrunculin B in the legend of Figure EV3B (line 521).

We maintain that the combined evidence of figures 2A and EV3A-C supports the conclusion that “*These findings suggest that LN411 induces endocrinogenesis via a reduction in cell-ECM adhesion*” (lines 113-114), especially in light of the further evidence of figure 2B-D presented later in the section.

4) *The knockdown experiments of ITGA6 raise some concerns, mostly due to the minor effect of the targeting siRNA (see Fig. 2), which did not result into a statistically significant downregulation of ITGA6. Perhaps, CRISPR interference (CRISPRi) would be a more robust approach. Also, do the two integrins ITGA6 and ITGA3 compensate for each other in a human context? Finally, what are the consequences on the expression of ITGB1, the most abundant subunit in this epithelium, by the different laminin isoforms and upon ITGA3/6 knockdown?*

Response: Although the knockdown efficiency of ITGA6 was low, it consistently reduced ITGA6 expression across independent replicates and significantly promoted NGN3 expression in a reproducible manner. Given the consistency of these results across experiments, we consider the siRNA-mediated ITGA6 knockdown findings to be reliable.

The idea of ITGA3 and ITGA6 compensating for each other in human cells has not been established. ITGA3 and ITGA6 are not exact substitutes for each other; they can exhibit overlapping functions and potentially compensate for each other's absence in certain contexts, particularly in the context of cancer cell behavior.

In addition, given our findings that LN411 induces endocrine specification of pancreatic progenitors, our study primarily focused on the mechanism by which LN411 promotes endocrinogenesis through reduced cell-ECM adhesion. Therefore, we concentrated on comparing fibronectin-binding and laminin-binding integrins. Since ITGB1 itself is not specific to either group, we did not include it in our analysis.

5) *The analysis of the mouse Itga3/a6 knockout pancreas should be completed and the analysis of the phenotype carried out in vivo as much as possible. The authors claimed that this was not possible because of embryonic lethality but it is unclear at which stage and why/how this would hamper the dissection of the embryonic pancreata and their analysis. Anyway, the very preliminary analysis carried out on explants suggested an expansion of the endocrine compartment when compared to the pancreatic epithelium.*

The authors conclude that this might be due to accelerated endocrine specification and defective duct lineage, but this has not been tested here.

Response: Both *Itga3*-null and *Itga6*-null mutations are recessively lethal at birth. Double mutant embryos are growth-retarded, particularly at E16.5, and exhibited multiple defects (De Arcangelis et al, 1999). To avoid capturing indirect influences by defective non-pancreatic tissues, we decided to study early (E11.5) pancreatic explants.

The data show that an expansion of the endocrine compartment occurs when compared to the pancreatic epithelium. The fact that duct development was compromised suggests an increased conversion of bipotent progenitors towards the endocrine lineage at the expense of the duct lineage.

6) The role of PDK4 in endocrinogenesis is novel and interesting, however some of the findings in support of this conclusion are still preliminary. First, the analysis of the knockdown experiments using siRNA targeting PDK4 in hESCs should be expanded. For example, the requirement of PDK4 for PDX1 maintenance or for the regulation NKX6.1 and YAP should be tested in the PDK4-depleted cells instead of upon treatment with DCA or Acetate. These compounds exert multiple effects on pathways of intermediary metabolism and, therefore, have stronger and broader effects than just only on PDK4, possibly acting via other mechanisms.

Response: DCA affects PDK1-3 in addition to PDK4, while acetate affects the downstream levels of acetyl-CoA, which are affected by a multitude of pathways in addition to the activity of PDK1-4. While we agree with the reviewer that the specificity of PDK4-targeting siRNA is preferable, the small effect size achievable in our system unfortunately precludes the use of this method in place of the more potent DCA and acetate treatments.

However, we believe that the current evidence strongly supports the conclusion of PDK4-specificity. First, the consistent results of 3 different modalities for affecting PDK4-activity all converge on endocrine induction (Figure 4C-E, 6A-B). Second, PDK4-upregulation precedes PDX1-upregulation/maintenance (in LN411 relative to FN) and upregulation of endocrine markers (Figure 4A). Finally, the canonical/non-canonical dual pathways are supported by the differential effects of targeting the downstream acetyl-CoA generation (affects PDX1 but not YAP) and targeting PDK4-activity directly (affects both PDX1 and YAP) (Figure 5A-C, 6C-F).

We have further strengthened this conclusion by adding measurement of acetyl-CoA upon exposure to FN/LN411 alone or alongside DCA/Acetate (new Figure EV6E). This data supports our findings by demonstrating the expected pattern of reduced

acetyl-CoA upon LN411 exposure (relative to FN), which is counteracted by DCA or acetate treatment. This pattern agrees with the expression level of PDX1 under the same conditions (Figures 4A, 6F).

Figure EV6. Acetate inhibits endocrine specification via a YAP-independent mechanism.

(E) ELISA analysis of intracellular acetyl-CoA concentrations in pancreatic progenitor cells at 3, 6, and 24 hours after reseeding on FN, LN411, or LN411 with 10 mM DCA or 20 mM sodium acetate. The data were analyzed by two-way repeated measures ANOVA testing for treatment differences across the entire time course (3-24 h), followed by Dunnett's test with comparison to LN411 alone and are shown as mean expression \pm SEM (n = 5).

Reviewer 2

This is an interesting study that investigates the early cues required for specification of endocrine progenitors in the developing pancreas (the precursors of insulin-producing β -islet cells). It is known that expression of the transcription factor Neurog3 plays a role in this process, which activates a Notch signalling pathway. Data for the authors' group further suggest involvement of YAP (upstream of Notch) and, hence, a mechano-transduction component. The authors here investigate the possibility that the ECM environment orchestrates these pathways, potentially providing the mechano-transduction component, through a novel PDK4 - pathway.

The focus is on laminins, the reason for which is not explained in the introduction but is presumably because laminins have long been implicated in insulin secretion by β -islet cells; human iPSC derived pancreatic endocrine progenitors are employed in in vitro studies.

The authors propose that specifically laminin 411 provides an early 'soft matrix' signal required for differentiation to endocrine progenitors, by down regulating integrin receptor that would normally promote adhesion to RGD containing matrices and some laminins and promoting expression of PDK4 and altering the metabolic status of the cells. The concept is interesting but the data provided do not support this hypothesis and several important controls are missing.

Critically, from the data provided it is not clear whether a laminin 411 containing basement membrane is ever in contact with the endocrine progenitors in vivo, nor whether simply laminin 411 reflects a soft matrix. Changes in mechanical signals provided by the ECM may indeed induced endocrine progenitor formation, however, whether laminin 411 provides this signal is not substantiated by the data shown, nor is the link between integrin expression changes and laminin 411 clear. The manuscript is therefore too preliminary for publication in EMBO J.

Major points

Overall in vitro experiments have a small number of replicates.

Response: All in vitro experiments were performed with at least three independent replicates and were subjected to statistical analysis. We believe the number of replicates is sufficient to support our conclusions.

One point that is not clear is whether the 9 laminin isoforms tested in Fig. 1 actually occur at the site of endocrine progenitor localization in vivo. From the data shown, it seems that only laminin a1 occurs in the ductal basement membrane, at least at the

one time point shown. Conclusions cannot be made from the data provided for laminin a2, a4 or a5, as resolution is poor and magnifications are too low. Also, different developmental stages, corresponding to commitment of endocrine progenitors, are not shown. This is important information as from the model proposed, at some early time point in differentiation the non-committed progenitors cells must come in contact with laminin 411, lose certain integrins and upregulate NGN3 and PDK4. Similarly, does in vivo integrin expression on progenitor cells show down regulation of integrins a5, av, a3 and a6 coincident with their interaction with laminin 411?

Response: Due to the highly dynamic nature of the developing pancreatic epithelium (Nyeng *et al*, 2019), a complete assessment of spatiotemporal ECM-cell interactions would require advanced live cell imaging with multi-colour mouse reporter lines. Static images could be misleading as pancreatic progenitors in general and NEUROG3+ cells in particular undergo significant and rapid cell migration. However, we have extensively mined available scRNA-seq datasets to strengthen the in vivo relevance of our findings:

Human fetal pancreas analysis: We analyzed published scRNA-seq data focusing on 8-10 weeks post-conception (wpc), which corresponds to the onset of human endocrinogenesis. We specifically chose this timepoint because:

- It contains well-defined bipotent progenitor and endocrine progenitor populations
- Later timepoints lack clear intermediate endocrine-specified cells and defined bipotent progenitors

Our analysis (new Figure EV8A) reveals that the expression pattern perfectly aligns with our model:

Figure EV8. Single cell RNA sequencing datasets validate the LN411-PDK4-endocrine specification pathway.

- (A) Log2 fold change in gene expression between endocrine progenitors (EP) and bi-potent trunk progenitors (trunk) at 8-10wpc in the OMIX001616 dataset

(Ma et al, 2023). p-values are derived from MAST analysis (Finak et al, 2015) followed by Benjamini-Hochberg correction for multiple comparisons.

Specifically:

- ITGA3, ITGA6, ITGA5, and ITGAV are all significantly downregulated
- PDK4 shows upregulation (though not reaching statistical significance, likely due to detection in a small fraction of cells)
- YAP target genes show the expected downregulation
- PDX1 is significantly upregulated in endocrine progenitors

The authors claim that laminin 411 results in reduced endocrine precursor cell adhesion- but this is not shown.

Response: The lower adhesion properties of LN411 compared to other laminin isoforms have been demonstrated in published studies (Ishikawa et al, 2014). Therefore, we did not repeat this experiment in the current study.

It is not clear to this reviewer, what the significance of the downregulation of α_v , α_5 , α_3 and α_6 is. Do the authors propose that integrin α_6 and α_3 mediate binding to laminin 411 or to other laminins which may signal other signals preventing endocrine progenitor differentiation? This could be tested in adhesion/inhibition assays. The authors also fail to address another major laminin α_4 receptor, MCAM, which mediates low affinity binding to this isoform only. What is its expression pattern during formation of endocrine progenitors, relative to laminin α_4 expression? Are other major non-integrin receptors expression altered in the course of endocrine progenitor commitment, eg Lutheran (BCAM) and dystroglycan (the latter of which has also been implicated in events upstream of Notch signalling) ?

Response: We previously demonstrated that mechanical tension governs the cell fate decision of bipotent pancreatic progenitors towards either an endocrine or duct fate (Mamidi et al, 2018). Here, we identify reduced cell-ECM adhesion as a major inducer of endocrinogenesis. Unexpectedly, we find that LN411 causes reduced cell-ECM adhesion (reduced mechanical tension) through a rapid global reduction of the transcription and surface expression of multiple integrins. Thus, it is a broad lowering of integrin expression rather than targeting of specific integrin receptor-ligand signalling that triggers a mechanical cellular state conducive to endocrinogenesis.

A caveat is that the laminin a4 knockout mouse, which was generated over 20 years ago, has no described pancreatic defect, although early stages of pancreas development have not been studied/reported. The absence of an overt pancreas effect in the Lama4-/- mouse contrasts with the data reported in the manuscript (especially that of the integrin knockouts, if indeed they should represent laminin receptors). Additionally, although the authors claim that integrin a3/a6 double knockouts do not have an overt morphological defect in pancreas anlage, this is not reflected in the data shown, where ductal formation seems severely altered and the organ smaller. Further, it is important to show whether duct lineage markers are in fact being downregulated over the upregulation of endocrine lineage markers.

Response: As mentioned above, it reduces mechanical tension through lowered cell-ECM adhesion, which triggers endocrinogenesis. The absence of an overt pancreatic defect in the Lama4 KO mouse may be attributed to the overlapping functions of other low-adhesion Laminins, such as LN211 (see Fig. 1A-D).

We do NOT claim that we find that Itga3/a6 double knockouts do not have an overt morphological defect. Instead, we state: “the pancreas of the Itga3/Itga6 double knockouts exhibited an expansion of the endocrine compartment, coupled with hypoplasia and reduced branching (Fig. 3B-D)” (see Results).

Specific comments regarding figs:

Fig 1 shows expression of glucagon and C-peptide by pancreatic progenitor cells plated on different laminins isoforms after 9 days of culture. However, the expression of these proteins before contact with laminins is not shown. This is important as the authors state in methods that the cells were maintained on laminin 521 prior to the experiment. Were the initial levels of glucagon and C-peptide similar among groups before plated on different laminins?

Response: We seeded hESC-derived pancreatic progenitors onto different laminin isoforms and assessed glucagon and C-peptide expression after 9 additional days of differentiation. At the pancreatic progenitor stage, prior to seeding, these cells did not express glucagon or C-peptide.

The authors further confirmed the endocrine commitment of cells by qPCR. Although levels of NEUROG3 (used here as the major indicator of endocrine specification) was higher on cells plated on LM411 than on FN, it was only about 0.7 higher. To substantiate the data, qPCR showing downregulation of genes related to maintaining progenitor cells in an undifferentiated state, i.e., Hes1 (Notch signaling), would be important.

Response: NEUROG3 expression is transient, so the 0.7-fold increase detected by qPCR reflects a momentary difference rather than a cumulative effect. Importantly, compared to FN, NEUROG3 expression is consistently and significantly higher when cells are plated on LN411. In addition, the expression levels of INSULIN and GLUCAGON are also significantly elevated on LN411 compared to FN. Together, these results indicate that LN411 promotes endocrine commitment.

Addition points

The authors should consider that integrins alpha 5 and alpha v can also bind to laminin 511 - this has been shown by the researcher who has provided the laminin antibodies to the authors. There is also no convincing data that integrin a3b1 can bind to laminin 411. This fact should be taken into consideration (or indeed tested in in vitro adhesion assays using the cells employed in the study).

Response: We appreciate this information. However, we believe that referring to integrins $\alpha 5$ and αV as “fibronectin-binding integrins” is justified based on the general literature and the marginal nature of integrin $\alpha 5$ and αV binding to LN511, as described for instance in (Russo et al, 2016).

In any case, we believe these concerns are marginal, as we do not claim that the signalling from any specific integrin-laminin pairing results in the observed effect. We discuss these pairings mainly as an expansion on previously published data indicating a relationship between integrin $\alpha 5$ -fibronectin binding as a key inhibitor of pancreatic endocrine fate choice – a picture that our expanded data shows is too simplistic. Rather, exposure to LN411 triggers a general downregulation of cell-ECM adhesion, as has previously been demonstrated in cancer cells (Takkunen et al, 2008).

Fig 3/ Supplementary fig. EV4 The authors showed in vivo expression of laminin alpha4 and endocrinogenesis. The immunofluorescence images shown in Fig. EV4 are inconclusive and do not support the statement "which revealed a heterogenous distribution of laminin $\alpha 4$ " (line 132-133). In fact, laminin alpha4 was previously described to be continuously expressed on the vasculature basement membranes (Korpos et al, Diabetes 62:531-542, 2013).

Please specify the scale bar sizes in the legend or increase the size of those in the images.

Response: The study the reviewer refers to relates to expression in the vascular basement membrane in the adult pancreas. Proteomics data have shown that

LAMA4 maintains relatively high expression in the fetal pancreas compared to the adult pancreas. Additionally, in the fetal pancreas, LAMA4 exhibits an islet-to-acinar ratio of almost two. This further supports the dynamic expression of this subunit according to the endocrine developmental needs (Li et al, 2021).

We have updated the scale bar sizes of Figure EV4 to 100µm and specify the new scale bar size in the legend (line 535).

The authors should revisit the discussion section, where some references are not correct; for instance, authors refer to "Mahmoud, 2023" (a review article) when discussing cell fate decisions and cell-ECM interactions (lines 255-256), where the experimental paper that first described such interactions is important to cite.

Response: The point that original research should be cited even when referring to a general literature is well-taken. For this reason, we cite two recent relevant papers alongside the review article.

Furthermore, the downregulation of integrins by laminin 411 coating is vaguely discussed. Such an event, presented as an important part of the manuscript, should be better discussed. Does this down regulation of integrins lead to loss of adhesion of the cells to the substrate; is loss of adhesion per se a signal for differentiation of endocrine progenitors. In the discussion, it would be beneficial to address whether the downregulation of integrins by cells upon contact with laminins was reported previously, in particular in other endocrine systems.

The authors could consider further discussion on pancreas blood vessels (BV) and its involvement in the the development of the endocrine pancreas. The function of BVs in pancreas is mentioned in the discussion and that the literature so far is rather unclear, however, for this report in which BV basement membranes probably contribute substantially to the events presented, it's potential influence could be discussed.

Response: As expected, adhesion substrate adhesion to low adhesion LNs, such as LN411 and LN211, resulted in more a more multi-layered 3D cell architecture. This is a common phenomenon when cells increase their cell-cell adhesion upon reduced adhesion to their substrate.

During development, LN411 is known to be secreted by a subset of endothelial cells and by Schwann cells in the developing pancreas (Olaniru et al, 2023; Frieser et al, 1997). Single cell analysis of human pancreatic development confirms that both Schwann cells and endothelial cells express LAMA4 at high levels during the onset of endocrinogenesis at 8-10 wpc (Olaniru et al, 2023). Schwann cells have recently been shown to co-localize with endocrine progenitor cells during human pancreas

development (Olaniru *et al*, 2023). Whether Schwann cells play a functional role during pancreas development remains to be investigated. Existing data regarding the role of endothelial cells in endocrinogenesis is conflicting. While hypervascularization and endothelial cell depletion studies demonstrated a positive influence on endocrinogenesis by endothelial cells (Lammert *et al*, 2001; Pierreux *et al*, 2010), other studies identified mechanisms by which endothelial cells inhibit endocrinogenesis (Magenheim *et al*, 2011; Kao *et al*, 2015). Discussion has been revised accordingly.

We are unaware of other examples where downregulation of integrins affects cell fate in other endocrine systems.

Authors mix gene names with protein names- Lama4, Lama1, etc are gene names, written in Italics, the proteins are laminin a4, laminin a1 etc. This should be corrected in figures and the text, as it leads to confusion.

Response: We found a few cases of inconsistent nomenclature, which we have addressed. In particular, we are now correctly italicizing names that could refer equally well to gene or protein names, and we have updated the text on figure EV4. To follow the correct nomenclature.

Statistics are missing in some graphs from Fig 2B.

Response: These are because we consistently only show p-values below 0.05, as stated in the methods section.

The focus is on laminins, the reason for which is not explained in the introduction but is presumably because laminins have long been implicated in insulin secretion by b-islet cells; human iPSC derived pancreatic endocrine progenitors are employed in in vitro studies.

Response: In a previous study, we identified reduced mechanical tension as a key event in endocrine induction (Mamidi *et al*, 2018). As expected, different ECM components induces different degrees of mechanical tension, with fibronectin inducing the most tension (and endocrinogenesis) while laminin derived from human placenta (with unknown isoform composition) supported the highest levels of endocrinogenesis.

Reviewer 3

In this manuscript, Ebeid et al. investigate how extracellular matrix (ECM) cues, specifically low-adhesion laminin 411 (LN411), influence pancreatic endocrine lineage allocation. The authors propose that LN411 reduces integrin $\alpha 3$ and $\alpha 6$ expression, thereby lowering cell-ECM adhesion and mechanical tension, which leads to upregulation of PDK4. PDK4 is suggested to promote endocrine specification through two distinct mechanisms: (1) non-canonical inhibition of YAP signaling and (2) canonical reduction of acetyl-CoA to sustain PDX1 expression.

The study addresses a significant biological question at the intersection of ECM signaling, mechanotransduction, metabolism, and endocrine fate choice, with potential relevance for understanding pancreas development and improving in vitro differentiation protocols for diabetes therapy.

While the study presents interesting and novel connections, particularly the involvement of PDK4 as a mechanosensitive metabolic regulator, several issues limit the strength and novelty of the conclusions. The mechanistic pathways are not fully developed, and in vivo validation is limited, raising concerns about the physiological relevance of the proposed mechanism. Furthermore, some aspects of the manuscript are not well articulated, and the focus on PDK4 may overlook other relevant pathways suggested by the authors' RNA-seq data.

Overall, while the study has potential value, I believe major revisions are needed to fully support the conclusions and to improve the clarity and impact of the manuscript.

Major Concerns

Incomplete mechanistic understanding of PDK4's dual role (YAP inhibition and PDX1 maintenance):

The proposed non-canonical inhibition of YAP by PDK4 is central to the model but lacks mechanistic evidence. It remains unclear whether this occurs through known intermediates (e.g., Septin-2) or other unknown substrates. Similarly, the claim that PDK4-mediated reduction of acetyl-CoA maintains PDX1 expression is speculative and lacks direct metabolite measurements (e.g., acetyl-CoA levels, FOXA2 acetylation status). The authors should provide additional experimental data, or at least address these gaps clearly in the discussion and propose a mechanistic hypothesis based on existing literature.

Response: We have updated Figure 4A to include PDX1 expression data and revised the text (lines 188-190 and 225-227) to emphasize the temporal patterns revealed by our time-course RNA sequencing. We now highlight the coordinated upregulation of

PDK4 and PDX1 alongside downregulation of YAP targets CCN1 and CCN2 at the 6-hour time point, which precedes significant upregulation of endocrine specification genes, including NEUROG3.

Figure 4. LN411 induces endocrine specification via PDK4

(A) Volcano plots of bulk RNA-seq analysis of NGN3-GFP-derived pancreatic progenitors reseeded on LN411 or FN at indicated time points. Expression of YAP target genes (CCN1/CYR61, CCN2/CTGF), early endocrine genes (NEUROG3, NEUROD1, INSM1, NKX2-2), PDX1 and PDK4 are indicated.

In addition, we have added new experimental data analyzing acetyl-CoA dynamics upon exposure to FN, LN411 alone or LN411 in combination with DCA or acetate (new Figure EV6E, lines 227-230).

These data directly demonstrate:

- Reduced intracellular acetyl-CoA levels upon LN411 exposure (as compared to FN)
- The LN411-induced reduction is counteracted by concurrent DCA or acetate treatment
- The LN411-induced reduction of intracellular acetyl-CoA precedes the observed downregulation of PDX1 at 48 hours

These temporal relationships strongly support our model, which suggests that the PDK4-mediated reduction of acetyl-CoA maintains PDX1 expression during endocrine specification.

Figure EV6. Acetate inhibits endocrine specification via a YAP-independent mechanism.

(E) ELISA analysis of intracellular acetyl-CoA concentrations in pancreatic progenitor cells at 3, 6, and 24 hours after reseeding on FN, LN411, or LN411 with 10 mM DCA or 20 mM sodium acetate. The data were analyzed by two-way repeated measures ANOVA testing for treatment differences across the entire time course (3-24 h), followed by Dunnett's test with comparison to LN411 alone and are shown as mean expression \pm SEM (n = 5).

Limited in vivo validation and physiological relevance:

The study relies primarily on in vitro systems and ex vivo explants, with no direct in vivo evidence of LN411's role in endocrine specification. The spatial relationship between LN411 deposition and NEUROG3+ progenitors during pancreas development is not clearly shown. Additional data (e.g., in vivo co-localization of LN411 and endocrine progenitors, conditional models) or a more thorough discussion of these limitations is necessary to validate the proposed mechanism in a physiological context.

Response: Using the comprehensive (Olaniru et al, 2023) dataset, we assessed LAMA subtype expression across human pancreatic development (new Figure EV8B):

Figure EV8. Single cell RNA sequencing datasets validate the LN411-PDK4-endocrine specification pathway.

(B) Dot plots of normalized expression of laminin α subunit genes in single-cell RNA sequencing samples of fetal pancreata according to developmental time (left) and cell type (right). Plots generated using humanpancreasdevelopment.org (Olaniru et al, 2023).

This analysis specifically shows that:

- LAMA4 expression peaks at 8-10 wpc, precisely coinciding with endocrine specification onset.
- LAMA4 is predominantly expressed by non-epithelial cells, particularly Schwann cells and endothelial cells.
- This pattern supports our model of heterogeneous LN411 deposition from surrounding mesenchyme driving local endocrine fate decisions

We have added these findings on lines 235-256 and 286-299 in the discussion, providing in vivo expression patterns validation of our proposed LN411-PDK4 pathway in human pancreatic development.

Selective focus on PDK4 without addressing other candidates from RNA-seq data: The RNA-seq analysis identifies other significantly upregulated genes (e.g., FOXA3, RFX6, FOXP1, DDIT4), some of which are known to influence pancreatic development. The authors should discuss these additional factors, and whether endocrine induction may involve a multifactorial response rather than a PDK4-centric pathway. This would provide a more balanced interpretation of their findings.

Response: We have revised lines 174-183 to more clearly articulate why PDK4 was selected from the 9 overlapping genes between LN411 and latrunculin B treatments. We now explicitly state that the known functions of FST, RFX6, RGS4, and FOXP1 make them unlikely candidates for mediating the endocrinogenic effects. Specifically:

- FST has been shown to downregulate rather than upregulate endocrine induction (Miralles et al, 1998)
- RFX6, RGS4, and FOXP1 function downstream of NEUROG3 in endocrine development, delamination, and α -cell function, respectively (Serafimidis et al, 2011; Smith et al, 2010; Spaeth et al, 2015)
- In contrast, DDIT4, MAN1A1, and PDK4 have not been previously implicated in pancreas development, while PDK4's established role in ECM attachment

(Grassian *et al*, 2011) and anchorage-independent growth (Tambe *et al*, 2019) makes it a compelling candidate for further investigation.

Articulation and clarity of the proposed model: The manuscript contains dense and sometimes confusing descriptions of mechanisms, especially regarding the dual role of PDK4. The authors should revise key sections for clarity and logical flow, and clearly define terms such as "canonical" and "non-canonical" early in the manuscript. Including a schematic summary figure of the proposed mechanism would help guide the reader.

Response: We have modified the initial discussion of the link between PDK4 and PDX1 to reflect the updated figure 4A and new figure EV6E. We further provide a schematic of the proposed pathway in new figure EV7:

Figure EV7. PDK4 induces endocrine fate choice via both canonical and non-canonical functions.

Schematic illustrating the proposed method of PDK4-mediated endocrine fate choice. Canonically, PDK4 blocks conversion of pyruvate to acetyl-CoA via inhibition of PDH (not shown). PDK4 additionally blocks YAP activity through a non-canonical mechanism. PDX1 is required for *NGN3* expression, while YAP inhibits *NGN3* expression. DCA targets PDK4 directly and thus increases YAP activity and decreases *PDX1* expression. Acetate is converted to Acetyl-CoA, bypassing the effect of PDK4. Acetate treatment thus reduces *PDX1* expression without affecting YAP activity.

We hope that these changes are sufficient to provide clarity about the proposed mechanism.

Minor Concerns

Functional application in hPSC-derived beta-cell differentiation: Given the potential translational relevance, exploring whether LN411/PDK4 modulation could improve beta-cell differentiation protocols would enhance the impact of the study. Even preliminary data could provide an interesting proof-of-concept.

Response: The data in Figures 1A-D show that reseeding hPSC-derived pancreatic progenitors specifically on LN411 is a way to promote differentiation of INS⁺ or CPEP⁺ beta cells.

To target PDK4 as a way to improve beta cell generation, one would need a PDK4 agonist, which we have been unable to identify.

Further exploration of PDK4's non-canonical effects: Investigating potential interacting proteins or signaling partners of PDK4 that may mediate YAP inhibition (e.g., via proteomics or proximity-labeling approaches) would add mechanistic depth.

Response: Further exploration of the non-canonical effects of PDK4 is indeed an interesting and worthwhile direction. However, we believe it is beyond the scope of the current study. In future work, we plan to investigate potential interacting proteins or signaling partners of PDK4 that may mediate YAP inhibition.

Consideration of temporal dynamics: Time-course experiments measuring acetyl-CoA, YAP localization, PDK4 expression, and NEUROG3 induction could clarify the sequence of events and strengthen the causality of the proposed pathway.

Response: We have comprehensively addressed the temporal dynamics as suggested:

Enhanced temporal analysis: We have updated Figure 4A to include PDX1 expression data and revised the text (lines 188-190 and 225-227) to emphasize the temporal patterns revealed by our time-course RNA sequencing.

Figure 4. LN411 induces endocrine specification via PDK4

(A) Volcano plots of bulk RNA-seq analysis of NGN3-GFP-derived pancreatic progenitors reseeded on LN411 or FN at indicated time points. Expression of YAP target genes (CCN1/CYR61, CCN2/CTGF), early endocrine genes (NEUROG3, NEUROD1, INSM1, NKX2-2), PDX1 and PDK4 are indicated.

We now highlight the coordinated upregulation of PDK4 and PDX1 alongside downregulation of YAP targets CCN1 and CCN2 at the 6-hour time point, which precedes significant upregulation of endocrine specification genes, including NEUROG3.

Acetyl-CoA measurements: We have added new experimental data analyzing acetyl-CoA dynamics upon exposure to FN, LN411 alone or LN411 in combination with DCA or acetate (new Figure EV6E, lines 227-230).

Figure EV6. Acetate inhibits endocrine specification via a YAP-independent mechanism.

(E) ELISA analysis of intracellular acetyl-CoA concentrations in pancreatic progenitor cells at 3, 6, and 24 hours after reseeding on FN, LN411, or LN411 with 10mM DCA or 20mM sodium acetate. The data were analyzed by two-way repeated measures ANOVA testing for treatment differences across the entire time course (3-24h), followed by Dunnett's test with comparison to LN411 alone and are shown as mean expression \pm SEM (n = 5).

These data directly demonstrate:

- **Reduced intracellular acetyl-CoA levels upon LN411 exposure (as compared to FN)**
- **The LN411-induced reduction is counteracted by concurrent DCA or acetate treatment**
- **The LN411-induced reduction of intracellular acetyl-CoA precedes the observed downregulation of PDX1 at 48 hours**

These temporal relationships strongly support our model, which suggests that the PDK4-mediated reduction of acetyl-CoA maintains PDX1 expression during endocrine specification.

Based on the above points, I recommend Major Revision if this manuscript is being considered for EMBO, as it addresses a novel and interesting aspect of pancreas development that could be suitable with significant clarification and additional data or discussion. However, in my view, this manuscript may not yet meet the standard for EMBO Journal, as it requires substantially deeper mechanistic and in vivo validation to reach the high bar of conceptual and technical advance expected there.

References

- De Arcangelis A, Mark M, Kreidberg J, Sorokin L & Georges-Labouesse E (1999) Synergistic activities of $\alpha 3$ and $\alpha 6$ integrins are required during apical ectodermal ridge formation and organogenesis in the mouse. *Development* 126: 3957–3968
- Finak G, McDavid A, Yajima M, Deng J, Gersuk V, Shalek AK, Slichter CK, Miller HW, McElrath MJ, Prlic M, *et al* (2015) MAST: A flexible statistical framework for assessing transcriptional changes and characterizing heterogeneity in single-cell RNA sequencing data. *Genome Biol* 16: 1–13
- Frieser M, Nöckel H, Pausch F, Röder C, Hahn A, Deutzmann R & Sorokin LM (1997) Cloning of the Mouse Laminin $\alpha 4$ cDNA. Expression in a Subset of Endothelium. *Eur J Biochem* 246: 727–735
- Grassian AR, Metallo CM, Coloff JL, Stephanopoulos G & Brugge JS (2011) Erk regulation of pyruvate dehydrogenase flux through PDK4 modulates cell proliferation. *Genes Dev* 25: 1716–1733
- Ishikawa T, Wondimu Z, Oikawa Y, Gentilcore G, Kiessling R, Egyhazi Brage S, Hansson J & Patarroyo M (2014) Laminins 411 and 421 differentially promote tumor cell migration via $\alpha 6\beta 1$ integrin and MCAM (CD146). *Matrix Biology* 38: 69–83
- Kao DI, Lacko LA, Ding B, Sen, Huang C, Phung K, Gu G, Rafii S, Stuhlmann H & Chen S (2015) Endothelial Cells Control Pancreatic Cell Fate at Defined Stages through EGFL7 Signaling. *Stem Cell Reports* 4: 181
- Lammert E, Cleaver O & Melton D (2001) Induction of pancreatic differentiation by signals from blood vessels. *Science* (1979) 294: 564–567
- Li Z, Tremmel DM, Ma F, Yu Q, Ma M, Delafield DG, Shi Y, Wang B, Mitchell SA, Feeney AK, *et al* (2021) Proteome-wide and matrisome-specific alterations during human pancreas development and maturation. *Nature Communications* 2021 12:1 12: 1–12
- Ma Z, Zhang X, Zhong W, Yi H, Chen X, Zhao Y, Ma Y, Song E & Xu T (2023) Deciphering early human pancreas development at the single-cell level. *Nature Communications* 2023 14:1 14: 1–17
- Magenheim J, Ilovich O, Lazarus A, Klochendler A, Ziv O, Werman R, Hija A, Cleaver O, Mishani E, Keshet E, *et al* (2011) Blood vessels restrain pancreas branching, differentiation and growth. *Development* 138: 4743–4752
- Mamidi A, Prawiro C, Seymour PA, de Lichtenberg KH, Jackson A, Serup P & Semb H (2018) Mechanosignalling via integrins directs fate decisions of pancreatic progenitors. *Nature* 2018 564:7734 564: 114–118

- Miralles F, Czernichow P & Scharfmann R (1998) Follistatin regulates the relative proportions of endocrine versus exocrine tissue during pancreatic development. *Development* 125: 1017–1024
- Nyeng P, Heilmann S, Löf-Öhlin ZM, Pettersson NF, Hermann FM, Reynolds AB & Semb H (2019) p120ctn-Mediated Organ Patterning Precedes and Determines Pancreatic Progenitor Fate. *Dev Cell* 49: 31-47.e9
- Olaniru OE, Kadolsky U, Kannambath S, Vaikkinen H, Fung K, Dhimi P & Persaud SJ (2023) Single-cell transcriptomic and spatial landscapes of the developing human pancreas. *Cell Metab* 35: 184-199.e5
- Pierreux CE, Cordi S, Hick AC, Achouri Y, Ruiz de Almodovar C, Prévot PP, Courtoy PJ, Carmeliet P & Lemaigre FP (2010) Epithelial: Endothelial cross-talk regulates exocrine differentiation in developing pancreas. *Dev Biol* 347: 216–227
- Russo J Di, Luik A, Yousif L, Budny S, Oberleithner H, Hofschroer V, Klingauf J, Bavel E van, Bakker EN, Hellstrand P, *et al* (2016) Endothelial basement membrane laminin 511 is essential for shear stress response. *EMBO J* 36: 183
- Serafimidis I, Heximer S, Beis D & Gavalas A (2011) G Protein-Coupled Receptor Signaling and Sphingosine-1-Phosphate Play a Phylogenetically Conserved Role in Endocrine Pancreas Morphogenesis. *Mol Cell Biol* 31: 4442–4453
- Smith SB, Qu HQ, Taleb N, Kishimoto NY, Scheel DW, Lu Y, Patch AM, Grabs R, Wang J, Lynn FC, *et al* (2010) Rfx6 directs islet formation and insulin production in mice and humans. *Nature* 2010 463:7282 463: 775–780
- Spaeth JM, Hunter CS, Bonatakis L, Guo M, French CA, Slack I, Hara M, Fisher SE, Ferrer J, Morrissey EE, *et al* (2015) The FOXP1, FOXP2 and FOXP4 transcription factors are required for islet alpha cell proliferation and function in mice. *Diabetologia* 58: 1836–1844
- Takkunen M, Ainola M, Vainionpää N, Grenman R, Patarroyo M, García De Herreros A, Konttinen YT & Virtanen I (2008) Epithelial-mesenchymal transition downregulates laminin α 5 chain and upregulates laminin α 4 chain in oral squamous carcinoma cells. *Histochem Cell Biol* 130: 509–525
- Tambe Y, Terado T, Kim CJ, Mukaisho K ichi, Yoshida S, Sugihara H, Tanaka H, Chida J, Kido H, Yamaji K, *et al* (2019) Antitumor activity of potent pyruvate dehydrogenase kinase 4 inhibitors from plants in pancreatic cancer. *Mol Carcinog* 58: 1726–1737

Dear Henrik,

Thank you once more for the submission of your revised manuscript to EMBO reports. As you know, we have received the full set of referee reports (copied again below). Referee #2 had remaining concerns, but since these pertain to issues we had not asked you to address experimentally, we will disregard these and proceed with publication. In fact, I had discussed this further with referee #2 and s/he agreed and also acknowledged that you discuss and acknowledge the limitation of in vivo relevance in the manuscript.

I have now finalized all checks from the editorial side. I am sorry that this took a bit longer, but we are currently not fully staffed.

These are the points we would need you to address before we can proceed with formal acceptance:

- Please remove the figures from the manuscript.
- Please place the main and EV figure legends at the end of the manuscript (sections Figure legends, Expanded View figure legends)
- Please provide up to 5 keywords.
- Materials and Methods should be Methods
- Studies involving experimental animals: please also provide the reference number for approval in the methods section.
- The scale bars in the figures are often quite thin, e.g. in Figure 1A. Moreover, please define the size only in the legends, not on the scale bar in the image. In the image itself the font is too small to be readable (see Fig. EV2C as an example).
- Figure 3B and the left images shown in Figure 3C need scale bars.
- Please place the Data Availability section before the Acknowledgments and please remove the statements "Numerical data underlying the graphs presented in the manuscript are provided in the source data files. All additional source data is available from the corresponding author upon reasonable request." as this section should only refer to datasets deposited in a public repository.
- The competing interests section needs to be renamed to "Disclosure and Competing Interests Statement" and the Ethics declarations needs to be removed
- Regarding the Author Contributions, we now use CRediT to specify the contributions of each author in the journal submission system. Therefore, please remove the Author Contributions from the manuscript file and make sure that the author contributions in our online manuscript tracking system are correct and up-to-date. The information you specified in the system will be automatically retrieved and typeset into the article. You can enter additional information in the free text box provided, if you wish.
- A callout for Fig. 4C appears to be missing in the manuscript text.
- Could you please provide a short (1-2 sentences) summary of the findings and their significance, and 2-3 bullet points highlighting key results? This text will accompany the synopsis summary image on the webpage.

Please address the following comments regarding figure legends:

- a) Please indicate the statistical test used for data analysis in the legends of figures 1B, D; 3D, 4A, EV5 A, EV7 A
- b) Please note that the box plots need to be defined in terms of minima, maxima, centre, bounds of box and whiskers, and percentile in the legends of figures 2A, EV3 C
- c) Please note that information related to n is missing in the legends of figures 4A, EV5 A"

- Finally, I introduced a few minor edits to the abstract (copied below). Could you please review it?

With kind regards,

Martina

Martina Rembold, PhD

Senior Editor
EMBO reports

=====

Referee #1:

The revised manuscript is significantly improved, compared with the last version. The authors have addressed my concerns and this paper should be published in Embo reports.

Referee #2:

Dear Editor,

Thank you for granting me additional time to this review process. I have evaluated the authors' rebuttal details, and the new data provided. While the additional analyses, specifically the acetyl-CoA dynamics (EV6E) and scRNA-seq mining, add more observations to their model, unfortunately, my primary concerns remain unresolved. Specifically, the mechanistic basis of the role of PDK4 in regulating YAP activity and maintaining PDX1 expression remains speculative at best. While the new data demonstrates temporal correlations, it still not not establishes causality nor clarify the intermediates involved. In addition, the physiological relevance of LN411 signaling in vivo remains unclear. The manuscript continues to rely heavily on its in vitro systems, and the absence of convincing in vivo evidence (including the Lama4 knockout for example) weakens the claim that LN411 is a decisive cue for endocrine specification. Furthermore, while the emphasis on PDK4 is interesting, the RNA-seq data likely implicate additional pathways that are not explored. The discussion of integrin downregulation and metabolic rewiring also remains vague, limiting the clarity and broader impact of the work.

In summary, although the revisions improve certain aspects, the key mechanistic and physiological gaps I raised are not fully addressed. I therefore believe that substantial additional work is required before the manuscript can meet the standards of EMBO J.

Referee #3:

Most of my comments have been addressed or argued in a reasonable manner.

=====

Suggested Abstract:

The mechanical and metabolic states of progenitor and stem cells are emerging as key regulators of cell fate decisions. Lineage specification of pancreatic endocrine cells is promoted by reduced mechanical tension in vitro, but the underlying mechanism is poorly understood. Here, we show that heterogeneously deposited low-adhesion extracellular matrix (ECM) components, such as the laminin isoform LN411, trigger a local "soft" environment by broadly reducing the expression of integrins. Mimicking this low-tension state by in vitro knockdown and in vivo gene targeting of the LN-binding integrins Itga3 and Itga6 reveal their importance in inducing endocrinogenesis. Unexpectedly, the cell responds to this change in tensile forces by engaging a major metabolic enzyme, PDK4, to execute the resulting cell fate decision. PDK4 achieves this through two distinct mechanisms: a non-canonical action controlling YAP activity and a canonical metabolic function maintaining PDX1 expression. In sum, we believe our findings have broad relevance for how local changes in mechanical tension governs cell behaviour in many developmental and disease contexts.

All editorial and formatting issues were resolved by the authors.

Prof. Henrik Semb
Helmholtz Zentrum Munich
Institute of Translational Stem Cell Research
Ingolstädter Landstraße 1
Oberschleißheim, Bayern 85764
Germany

Dear Henrik,

Thank you for sending the missing information about the replicates, which I have now added to the manuscript text. I am now very pleased to accept your manuscript for publication in the next available issue of EMBO reports. Thank you for your contribution to our journal.

Kind regards,

Martina
